# Excess weight is associated with neurological and neuropsychiatric symptoms in post-COVID-19 condition: A systematic review and meta-analysis

Débora Barbosa Ronca[1,2,3¤]*, Larissa Otaviano Mesquita[1], Dryelle Oliveira[4],
Ana Cláudia Morais Godoy Figueiredo[2,5], Jun Wen[6,7], Manshu Song[3‡],
Kênia Mara Baiocchi de Carvalho[1,4‡]

**1** Faculty of Health Sciences, Graduate Program of Public Health, University of Brasília, Brasília, Brazil,
**2** Health Department of Federal District, Brasília, Brazil, **3** School of Medical and Health Sciences, Edith Cowan University, Perth, Western Australia, Australia, **4** Faculty of Health Sciences, Graduate Program of Human Nutrition, University of Brasília, Brasília, Brazil, **5** Superior School of Health Sciences, Brasília, Brazil, **6** Faculty of Hospitality and Tourism Management, Macau University of Science and Technology, Macau SAR, China, **7** Faculty of Business and Law. Curtin University, Perth, Western Australia, Australia,

‡ Co-corresponding authors.
¤ Current address: School of Medical and Health Sciences, Edith Cowan University, Perth, Western Australia, Australia
* d.ronca@ecu.edu.au, deboraronca@gmail.com

## Abstract

### Background/purpose

Excess weight has been identified as a potential risk factor for post-COVID-19 condition (PCC). This systematic review and meta-analysis aimed to investigate whether excess weight is associated with the development or experience of neurological and neuropsychiatric symptoms in PCC.

### Methods

We conducted a comprehensive search of eight databases (PubMed, Embase, Scopus, Web of Science, VHL, Google Scholar, ProQuest, and medRxiv) for studies published up to July 2023. Studies were included if they assessed PCC symptoms in relation to nutritional status, specifically the development of neurological and neuropsychiatric symptoms more than 12 weeks post-infection. The analysis compared exposure and controls groups (excess weight *vs.* normal weight; obesity *vs.* non-obesity). Data were synthesized using a random-effects model.

### Results

Of the 10,122 abstracts screened, 18 studies (*n* = 139,091 adults) were included. These studies included 79,050 individuals with excess weight *vs* 57,926 normal-weight individuals and 30,694 individuals with obesity *vs* 107,612 non-obese

**Data availability statement:** All relevant data are within the paper and its Supporting Information files.

**Funding:** This research received a grant from the Research Support Program at the Superior School of Health Sciences, Brasília, Brazil, funded by the Health Sciences Teaching and Research Foundation (Grant and Acceptance Term No. 5/2020 – FEPECS/DE) (DBR). This study was also partially financed by the Coordenação de Aperfeiçoamento de Pessoal de Nível Superior, Brasil – Finance Code 001 (DBR). MS is supported by the Western Australian Future Health Research and Innovation Fund (Grant ID WANMA/Ideas2023-24/10). KMBC is supported by National Council for Scientific and Technological Development CNPq (Grant n. 302740/2022-8). The funders had no role in the study design, data collection and analysis, decision to publish, or preparation of the manuscript.

**Competing interests:** The authors have declared that no competing interests exist.

individuals. The presence of excess weight in PCC was significantly associated with persistent depression (RR = 1.21; 95% CI: 1.03–1.42), headache (OR = 1.23; 95% CI: 1.10–1.37), memory issues (RR = 1.43; 95% CI: 1.24–1.65), sleep disturbance (RR = 1.31; 95% CI: 1.16–1.48), and vertigo (RR = 1.21; 95% CI: 1.04–1.41). Obesity was significantly associated with persistent headache (OR = 1.45; 95% CI: 1.37–1.53), numbness (RR = 1.61; 95% CI: 1.46–1.78), smell disorder (OR = 1.16; 95% CI: 1.11–1.22), taste disorder (OR = 1.22; 95% CI: 1.08–1.38), and vertigo (RR = 1.44; 95% CI: 1.35–1.53).

## Conclusions

Excess weight, including overweight and obesity, is associated with experiencing neuro-symptoms related to PCC. Individuals with these conditions urgently need enhanced personalized care management in current post-pandemic context.

## Introduction

A growing body of evidence suggests that a subset of COVID-19 survivors develop persistent, debilitating symptoms and may face a long road to complete recovery [1–4]. These symptoms have been shown to affect multiple organ systems, as evidenced by respiratory, cardiovascular, neurological, and mental health manifestations. This emergent condition has several names: long-COVID; post-acute sequelae of SARS-CoV-2 syndrome; and, as per the World Health Organization (WHO), post-COVID-19 condition (PCC). Although PCC has varying definition in the literature, it is generally described as involving persistent symptoms or new symptom onset, typically 12 weeks from the acute phase of COVID-19 [5,6], presenting a novel challenge to healthcare systems [7]. Recently, it was defined as an infection-associated chronic condition that occurs after SARS-CoV-2 infection [8].

To date, no comprehensive effective treatment has been recognized for PCC, and the common strategies are based on symptoms relief [9,10]. Among PCC's range of symptoms, long-term COVID-19 neurological and neuropsychiatric manifestations are of particular interest given the higher incidence of nervous system related sequalae's after post-viral epidemics [11]. Prior reviews revealed that smell and taste disorders, headache, sleep disturbance, anxiety and depression were among the most commonly reported persistent symptoms in the general population. However, these studies only follow participants for a few weeks after SARS-CoV-2 infection and do not meet the criteria for PCC [12–15]. The severity of the COVID-19 acute phase has been associated with the development of long-term COVID-19 neuropsychiatric symptoms [16]. The body mass index (BMI) has been regarded as a potential risk factor for PCC, although the association of excess weight (EW) with the development of specific neurological and neuropsychiatric symptoms remains unclear [17,18]. Moreover, studies that reported the prevalence of persistent neurological and neuropsychiatric symptoms up to one year after COVID-19 onset

exhibited considerable heterogeneity and did not perform subgroups analyses according to nutritional status [2,19]. An increased number of PCC symptoms that longer persist may be experienced by individuals with EW and might be associated to worsen health and poor quality of life [20].

It is noteworthy that EW is a chronic suboptimal heath condition characterized by excessive fat deposits and is considered a global public health problem. It represents a major risk factor for other chronic conditions and encompasses overweight, a suboptimal body weight that represents a risk to health, and obesity, a metabolic disease [21,22]. A complex yet costly care of individuals with EW is required, with appropriate multidisciplinary, long-term support [21]. In 2020, the overweight and obesity pandemic collided with the COVID-19 pandemic. These conditions' joint negative effects have increased the incidence of related diseases, leading to potentially adverse clinical and social consequences [23–26]. Obesity was found to be associated with poor COVID-19 outcomes, such as hospital admissions, intensive care admissions, and lethality rates [27–29]. Although the COVID-19 pandemic has come under control, the multisystemic nature of this disease and its long-term impacts have yet to be elucidated [12,30].

Although there is literature on the most reported neurological and neuropsychiatric symptoms of PCC, few studies reported the risk of PCC symptoms according to nutritional status and no previous review has explored the association of EW and the development of specific neurological or neuropsychiatric symptoms among COVID-19 survivors. Elucidating the long-term neuro-outcomes of COVID-19, along with its suite of symptoms and high-risk populations, is urgently needed to facilitate the development of reliable and personalized care management strategies. Although a universal definition has been recently stablished by the National Academies of Sciences, Engineering, and Medicine and the terminology of Long COVID has been encouraged to improve communication [8], the present study applied the definition of Post-COVID-19 Condition as proposed by WHO [5], since it was the most common used terminology among included studies. Therefore, this systematic review addresses whether EW is associated with the development or experience of specific persistent neurological or neuropsychiatric symptoms among COVID-19 survivors.

## Materials and methods

This study followed the Preferred Reporting Items for Systematic Reviews and Meta-Analyses (PRISMA) 2020 guidelines [31]. It is a subset of broad research aimed at investigating the role of EW in the development of persistent symptoms of PCC. This review was registered at the International Prospective Register of Systematic Reviews (PROSPERO) (ID: CRD42023433234).

### Search strategy and eligibility criteria

Two authors (DBR and LOM) searched seven electronic databases (MEDLINE, EMBASE, SCOPUS, Web of Science, VHL, Google Scholar, ProQuest) and a preprint server (medRxiv) on 3 July 2023. The search strategy was reviewed by two reviewers according to the criteria of the Peer Review of Electronic Search Strategies (PRESS) checklist [32]. The search strategy included the following terms: ("long COVID-19" OR "post-acute covid 19 syndrome" OR "COVID-19" OR "SARS-CoV-2" OR "post-COVID-19 condition" OR "long hauler") AND ("excess of weight" OR "overweight" OR "obesity" OR "body mass index") AND ("signs and symptoms" OR "COVID-19 sequelae" OR "headache" OR "loss of smell" OR "cognitive dysfunction" OR "anxiety" OR "depression" OR "sleep disorder" OR "brain fog)." The full search strategy is detailed in supporting information (S1 Table). Additionally, some articles were hand-searched to identify potentially eligible studies which might not be electronically retrieved, including those published in 2024 (S2 Table).

Studies were considered eligible for inclusion if they (a) had observational designs; (b) primarily focused on adults; (c) classified focal populations based on the presence of excess weight/obesity *versus* normal weight/non-obesity, using either measured BMI or self-reported data; and (d) had a mean (or median) follow-up of at least 12 weeks after the acute phase of COVID-19. Survivors could be (a) hospitalized or non-hospitalized, (b) inpatient or outpatient, and (c) mixed population (hospitalized and outpatient) recruited in the community, outpatient clinic or heath care system.

We exclude editorials, clinical trials, reviews, opinions, books or book chapters, conference abstracts, case reports, and correspondence articles. Studies were also excluded if they included mostly pregnant or nursing women; evaluated the health effects of COVID-19; assessed excess weight as a risk factor for acute-phase severity or mortality; investigated clusters of symptoms rather than specific symptoms; or if they lacked a control group (i.e., COVID-19 survivors who did not report persistent symptoms). Additionally, studies were excluded if they evaluated long-term sequelae of COVID-19 in a specific population (e.g., individuals with specific comorbidities) to ensure that our analysis remained focused on the general population of COVID-19 survivors.

All included studies adhered to ethical guidelines and were approved by their respective ethical committees.

## Screening process, study selection, and data extraction

The same two authors (DBR and LOM) independently screened the titles and abstracts of studies obtained from database searches and removed duplicates. The Rayyan app [33], a semi-automation tool, was used to streamline the review process. This was followed by a full-text review of the retained articles to ensure they met eligibility criteria (S2 Table). Fig 1 presents the flowchart for the included studies.

The study population of interest comprised COVID-19 survivors. Exposure was defined as EW or obesity based on BMI measurement, data from electronic medical records, or self-reported information. Two exposure groups were defined according to WHO guidelines [34]: 1) an excess weight group (BMI ≥ 25 kg/m$^2$) and 2) an obesity group (BMI ≥ 30 kg/m$^2$). Both Asian and Caucasian populations were included in the analysis, with corresponding BMI cut-offs applied: for

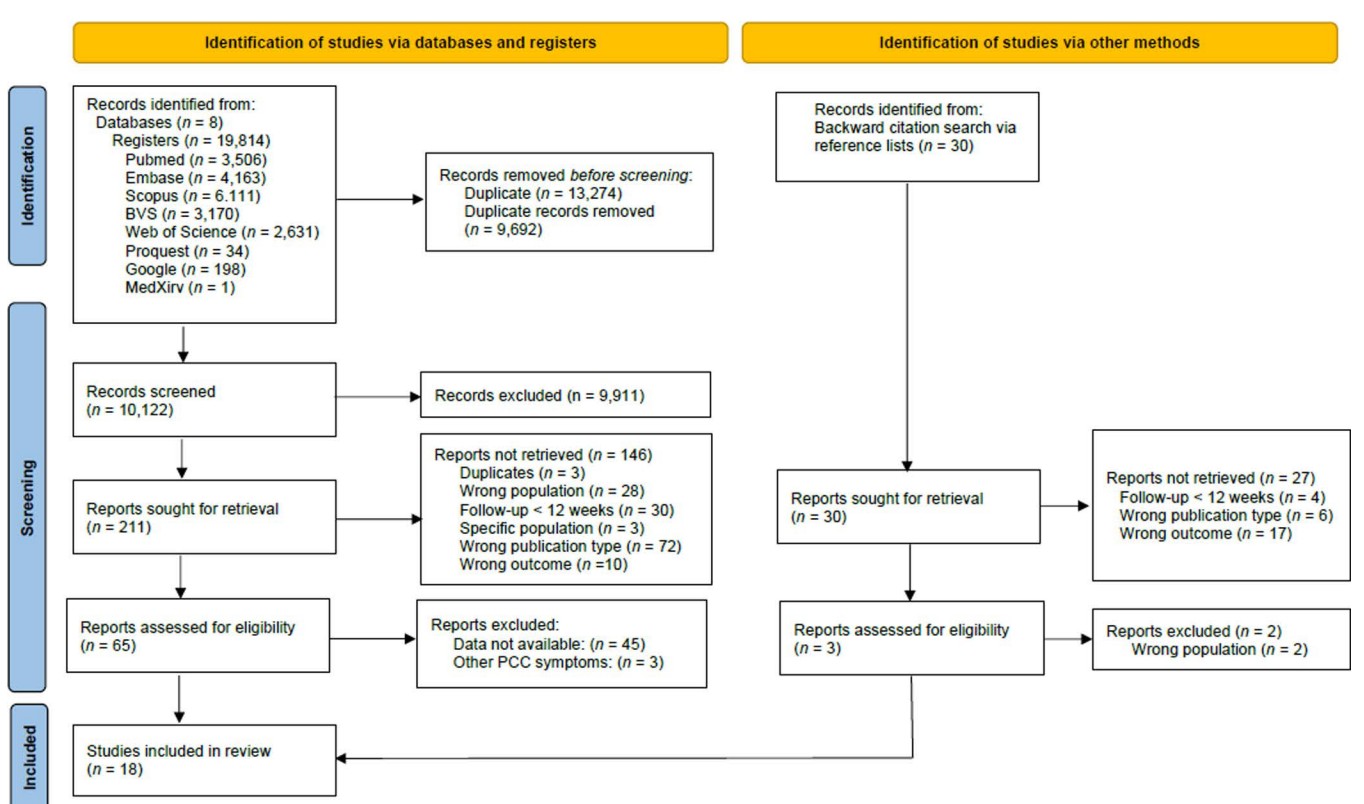

**Fig 1. Flow diagram of included studies.**

Asian-Pacific populations, the thresholds were BMI ≥ 23 kg/m² for EW and BMI ≥ 25 kg/m² for obesity, as previously defined in the included studies [35,36].

We adopted WHO's definition of PCC as outlined by Soriano *et al*. (2021) [5]. Symptoms were measured with pre-defined questionnaires/scales or reported during interviews (in person, by phone, or online). Due to varying definitions of illness onset (baseline), we accepted definitions that included 12 weeks from the onset of COVID-19 symptoms, COVID-19 diagnosis, hospital admission, or discharge after the acute phase. For the follow-up period, we considered studies where the sample median (or mean) of persistent symptoms was reported at least 12 weeks (84 days) after the baseline, with a minimum interquartile range (IQR) of 10 weeks (or a standard deviation (SD) of ± 24 days) from baseline. This cut-off was necessary to include studies evaluating long-term sequelae of COVID-19. Given the varied terminology used for symptoms across studies, we re-grouped symptoms into neurological and neuropsychiatric categories as shown in Fig 2.

Specific data were exported from each study into a predefined data collection form, including authors; publication year; country; aim of the study; study design; mean follow-up period; study population; exposure groups; assessment of outcome; evaluated symptoms; frequency of symptoms (*n*, %) among exposure and control groups; and the effect measure of the association of EW and the neurological/neuropsychiatric symptom. (S3 and S4 Table). Corresponding authors of relevant articles published were contacted if any data were absent (S5 Table).

## Quality assessment

The ROBINS-E tool for observational research was used to determine the risk of bias in the included studies [37]. ROBINS-E is a domain-based tool that evaluates seven domains (confounders, exposure measurement, participant selection, post-exposure interventions, missing data, outcome measurement, and selection of reported results) and then a general assessment of the study. Traffic plots with a final analysis of studies' risk of bias, by domain, is generated. For this research, two authors (DBR and DO) initially appraised the risk of bias. Disagreements were discussed and resolved in collaboration with a third researcher (LOM). The ROBINS-E tool was not applied as a study eligibility criterion.

We assessed the certainty of evidence via the Grading of Recommendations Assessment, Development and Evaluation (GRADE) framework [38–40] regarding the development of PCC-related neurological and neuropsychiatric symptoms

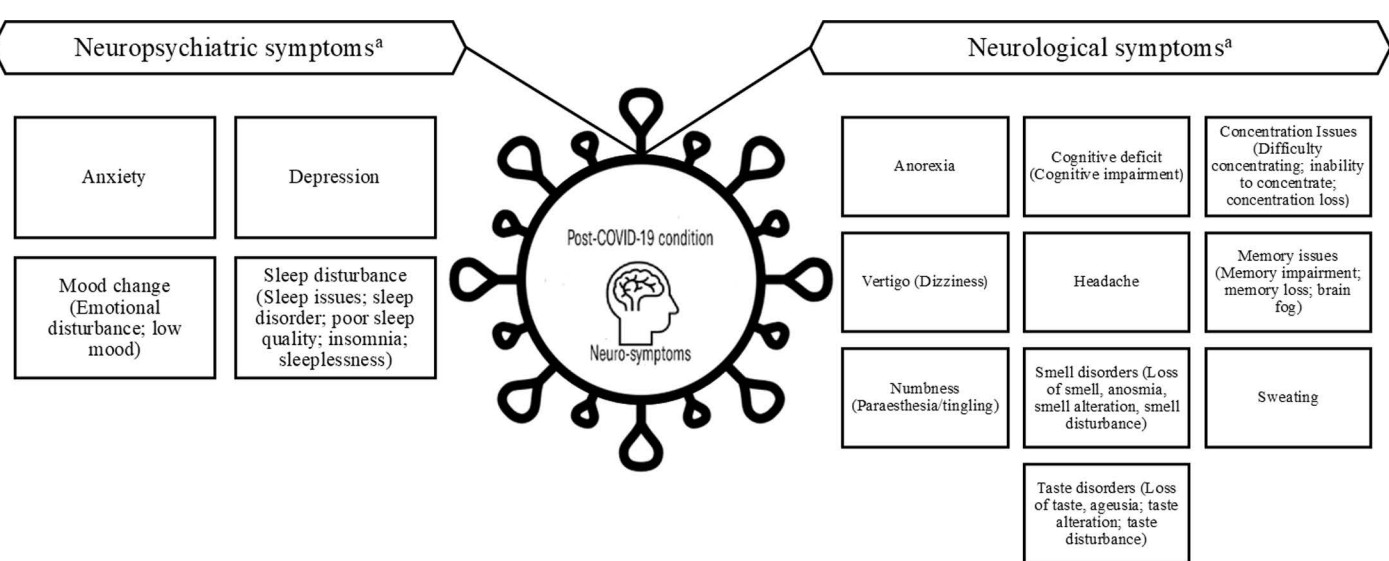

**Fig 2. Reported neurological and neuropsychiatric symptoms related to Post-COVID-19 condition (PCC).** [a] Neurological and neuropsychiatric symptoms reported in included studies were grouped into one category with synonyms presented in brackets.

among individuals with EW and obesity. The GRADE system measures key domains that impact the overall quality of evidence, including imprecision, inconsistency, indirectness, risk of bias, and publication (reporting) bias. The certainty of evidence was downgraded if these factors were identified. Moreover, the magnitude of the effect, dose-response gradient, and possible adjustment for confounding were also considered in the evaluation, which upgraded the quality of the evidence. Judgements for each GRADE domain were based on the information generated and synthesised in this systematic review. We assessed the certainty of evidence for all evaluated outcomes (symptoms) and scored it as follows: high certainty = ≥4 points (strong confidence that the true effect is close to the estimated effect); moderate certainty = 3 points (The true effect is likely to be close to the estimated effect, but there is some uncertainty); low certainty = 2 points (The true effect may be substantially different from the estimated effect); very low certainty = 1 point (The true effect is highly uncertain due to serious study limitations or inconsistencies).

## Statistical analysis

The frequency of reported symptoms and the associated risk of development were determined for EW *vs* normal weight groups and for obesity *vs* non-obesity groups, classified by BMI when available (normal weight: BMI of 18–24.9 kg/m$^2$; excess weight: BMI > 25 kg/m$^2$; obesity: BMI > 30 kg/m$^2$; non-obesity group: BMI < 30 kg/m$^2$), taking into account different cut-off values previously defined for the Asian population in studies [35,36]. Each symptom was considered an individual outcome. For this purpose, we collected data of the frequency (*n*, %) of each neurological and neuropsychiatric symptoms related to PCC reported and the risk of developing persistent symptoms (as measured by the odds ratio [OR] or the adjusted OR) at follow-up in included studies.

We performed meta-analytic calculations using STATA software (SE/17). Pooled risk ratios (RRs) with their 95% confidence intervals (CIs) were computed from the raw data of included cohort and cross-sectional studies, while pooled odds ratios (OR) were computed for reported symptoms identified in case-control studies. A random effects model meta-analysis was conducted to account for the statistical and methodological heterogeneity of the data. We used a two-sample binary-outcome summary dataset format with DerSimonian–Laird estimation when fewer than five studies were available [41]; when more than five studies were available, we adopted restricted maximum-likelihood estimation and to conduct pooled RR analysis. Statistical heterogeneity was assessed using the $I^2$ statistic, following thresholds recommended by Cochrane: 0%-40% (no important heterogeneity); 30%-60% (moderate heterogeneity); 50%-90% (substantial heterogeneity); and, 75–100% (considerable heterogeneity) [42]. Forest plots were generated for all outcomes. Publication bias was not assessed due to the limited number of included studies in each meta-analysis. A value of *p* < 0.05 was considered statistically significant.

## Results

Of the 10,122 abstracts screened, 211 full-text registers were reviewed, and 65 studies were considered eligible. Data on neurological and neuropsychiatric symptoms according to individuals' nutritional status were available for 18 studies (including one preprint [43]), all of which were entered into our meta-analysis.

Selected studies originated from 23 countries across Asia (China [35,36], India [44], Indonesia [45], Japan [46], Malaysia [43] and Saudi Arabia [47]), Europe (Denmark [48,49], England [50], Germany [51], Italy [52], Poland [53], Spain [54] and Switzerland [55]), and North and South America (Argentina [56], Brazil [56], Chile [56], Dominican Republic [56], Ecuador [56], Mexico [56], Panama [56], Paraguay [56], Peru [56] and United States of America [57,58]), comprising both developed and developing countries. Among the included studies, two employed case–control design [54,57], six involved retrospective or prospective cohorts [36,45,52,53,55,58], and 10 involved cross-sectional studies [35,43,44,47–51,56]. The populations investigated varied: three studies evaluated individuals in outpatient settings [49,55,57], five involved hospitalized patients [35,36,45,52,54], and ten included individuals either hospitalized or outpatient settings during COVID-19 acute-phase [43,44,46,47,50,51,53,56,58]. The classification of individuals according to nutritional status also varied: four

studies compared symptoms between individuals with EW and those with normal weight [35,43,45,47], five studies evaluated obesity *versus* non-obesity [36,51–54], eight studies presented data for individuals with normal weight, overweight and obesity [44,48–50,55–58], and one study only provided the mean BMI of the sample without specifying the number of individuals in exposure and control groups [46]. In total, our meta-analysis included 79,050 people with EW and 30,694 with obesity in exposure groups, while 57,926 normal weight individuals and 107,612 non-obese individuals comprised control groups. Sample sizes ranged from 32 [57] to 78,566 [50] COVID-19 survivors. Two Chinese studies applied different cut-off points for nutritional status classification (EW as BMI ≥ 23 kg/m$^2$; obesity as BMI ≥ 25 kg/m$^2$) [35,36]. Exposure was assessed via anthropometric measures [35,36,44,53,56,57], electronic medical record data [45,49,52,54], and self-reported comorbidity/BMI [43,46–48,50,55,58]. PCC neurological and neuropsychiatric symptoms were evaluated using questionnaires [36,44,46–50,53–55,57], validated scales [35,43,51,52,54,57], or self-report instruments [45,50,51,56,58]. The mean time from COVID-19 onset (baseline) to long-term symptom assessment was 25.8 weeks (range: 12 weeks [45,47,49,50,52] to 52 weeks [one year] [35]). Table 1 outlines the features of the included articles.

Half of the studies [35,36,43,46,49,53–55,58] tested for differences in the prevalence or risk of developing PCC neurological and neuropsychiatric symptoms based on nutritional status. One study found that BMI was inversely associated with the risk of developing smell and taste disorders [46]. Positive associations between EW and anxiety [35], depression [35,43], and sleep disturbance [54,55] were also reported, with obesity being independently associated with sleep disturbance [54]. Table 2 displays data on the frequency and risk of developing neurological and neuropsychiatric symptoms of PCC from studies that examined differences between exposure and control groups. Other studies [44,45,47,48,50–52,56,57] did not test for differences in the prevalence or risk of specific persistent neuropsychiatric symptoms. Data on symptom frequency by nutritional status are available in the supporting information (S6 Table).

## Excess weight and neuro-symptoms of PCC

We assessed the risk ratios (RRs) and odds ratios (ORs) for the EW group. Among neuropsychiatric symptoms, excess weight was significantly associated with persistent depression (RR = 1.21; 95% CI: 1.03–1.42; $I^2$ = 0.00) and sleep disturbance (RR = 1.31; 95% CI: 1.16–1.48; $I^2$ = 17.83%) (Fig 3). Moreover, significant positive association was observed between excess weight and headache (OR = 1.23; 95% CI: 1.10–1.17; $I^2$ = 40.36%), memory issues (RR = 1.43; 95% CI: 1.24–1.65; $I^2$ = 0.00), numbness (RR = 1.37; 95% CI: 1.24–1.51; $I^2$ = 0.00), and vertigo (RR = 1.21; 95% CI: 1.04–1.41; $I^2$ = 59.53%) (Fig 4A; Fig 4B and Fig 4C). Due to the limited number of studies, we were not able to perform meta-analysis for anorexia, mood change, concentration issues and swelling symptoms.

Four studies employed multivariate logistic regression and reported the risk of developing headache, smell disorder, memory impairment and taste disorder. These data were included in the meta-analysis. Pooled results revealed a significant association indicating that persistent taste disorder was inversely associated with BMI (OR 0.93; 95%CI 0.88–0.98; $I^2$ = 0,00). No significant associations were found for other evaluated symptoms (S1 Fig).

## Obesity and neuro-symptoms of PCC

Our pooled risk analysis for obesity and neurological symptoms (Fig 5A; Fig 5B and Fig 5C) showed that obesity was associated with a range of neurological symptoms: headache (OR = 1.45; 95% CI: 1.37–1.53; $I^2$ = 0.00); numbness (RR = 1.61; 95% CI: 1.46–1.78; $I^2$ = 0.00); smell disorder (0R = 1.16; 95% CI: 1.11–1.22; $I^2$ = 1.26%); taste disorder (0R = 1.22; 95% CI: 1.08–1.38; $I^2$ = 58.60%); and vertigo (RR = 1.44; 95% CI: 1.35–1.53; $I^2$ = 0.00). No statistically significant association was found between obesity and the risk of developing PCC-related neuropsychiatric symptoms (S2 Fig).

Substantial heterogeneity was observed in our meta-analysis for the non-significant association between obesity and anxiety ($I^2$ = 67.34%), while moderate heterogeneity was observed for sleep disturbance ($I^2$ = 51.27%), smell and taste disorder ($I^2$ = .51.43%) and taste disorder ($I^2$ = 58.60%) in pooled results comparing individuals with obesity and the control group. Meanwhile, moderate heterogeneity was observed when evaluating excess weight as the exposure group

**Table 1. Characteristics of included studies.**

| Author, Year | Country | Study aim | Study design; Follow-up[a] | Study population | Categories of nutritional status | Assessment type | Main neurological (N) and neuropsychiatric (NP) symptoms evaluated |
|---|---|---|---|---|---|---|---|
| Alkwai, H.M. et al., 2022 [47] | Saudi Arabia | To analyze the persistence of COVID-19 symptoms and return to the usual state of health | Cross-sectional; 12 weeks after COVID-19 infection | Mixed population[b] $n = 108$ (male = 23) Age range: 18–65 years | Overweight ($n = 32$) Normal weight ($n = 181$) | Online survey Self-reported comorbidities Symptom questionnaire | N: Concentration issues, headache, memory issues, numbness, smell disorder, taste disorder, vertigo NP: mood change, sleep disturbance |
| Blümel, J.E. et al., 2022 [56] | Nine Latin-American countries[c] | To study the development of long-term symptoms and the impacts of COVID-19 on mental health and quality of life in middle-aged women | Cross-sectional; 32 weeks after COVID-19 infection | Mixed population[b] Sample of interest/ study sample $n = 304/1,238$ (female only) Age: 53.0 (range: 40–64 years) | Obesity ($n = 41$) Overweight ($n = 124$) Normal weight ($n = 139$) | In-person interview Measured BMI Reported symptoms | N: anorexia, cognitive deficit, headache, memory issues, numbness, smell disorder, taste disorder, vertigo NP: anxiety, sleep disturbance |
| Bungenberg, J. et al., 2022 [51] | Germany | To identify and compare persistent self-reported symptoms in initially hospitalized and non-hospitalized patients after infection | Cross-sectional; 13.4 weeks for non-hosp. and 41 weeks for hosp. patients after acute symptoms | Mixed population[b] $n = 50$ (male = 22) Age: 50.5 (range: 40–64 years) | Obesity ($n = 7$) Non-obesity ($n = 43$) | In-person interview Reported symptoms and scales | N: cognitive deficit, memory issues, smell disorder, taste disorder, vertigo NP: sleep disturbance |
| Carter, S.J. et al., 2022 [57] | United States of America | To compare functional status, mood state, and physical activity in leisure time among positive COVID-19 and controls | Case–control; 12.1 weeks after COVID-19 diagnosis | Outpatient $n = 32$ woman SARS-CoV-2 group = 17 Age: 55 ± 11 years | Obesity ($n = 4$) Overweight ($n = 6$) Normal weight ($n = 7$) | In-person interview Measured BMI Questionnaire and Profile of Mood States | N: Cognitive deficit, headache, memory issues, smell disorder, taste disorder |
| Chudzik, M. et al., 2022 [53] | Poland | To analyse the prevalence of self-reported smell and/or taste disorders and to identify risk factors for the disease | Cohort; 28.8 weeks starting 14 days after last COVID-19 symptom | Mixed population[b] $n = 2,218$ (male = 1,410) Age: 53.8 ± 13.5 years | Obesity ($n = 692$) Non-obesity ($n = 1,485$) | In-person interview Measured BMI Health questionnaire | N: Smell and taste disorders |
| Desgranges, F. et al., 2022 [55] | Switzerland | To compare the prevalence of symptoms persisting for more than 3 months and to identify predictors of persistent symptoms | Cohort; 21.4 weeks after initial consultation (IQR: 17.2–29.1) | Outpatient Sample of interest / study sample $n = 418/507$ (male = 157) Age: 41 (range: 31–54 years) | Obesity ($n = 72$) Overweight/ obesity ($n = 189$) Healthy weight ($n = 229$) | Electronic medical records Telephone interviews Self-reported BMI Survey with predefined symptoms | N: Headache, loss of balance, memory issues, numbness, smell disorder, taste disorder NP: sleep disturbance |
| Epsi, N.J. et al., 2024 [58] | United States of America | To improve the definition of PCC with a data-driven approach to phenotyping. | Cohort 24 weeks post-infection | Mixed population[b] $n = 1,988$ (male = 1,201) Age: 69,5% were between 18–44 years old | Obesity ($n = 710$) Overweight ($n = 842$) Normal weight ($n = 436$) | Electronic patient records and self-reported BMI Online survey Survey with predefined symptoms | N: Smell and taste disorder |

*(Continued)*

**Table 1.** (Continued)

| Author, Year | Country | Study aim | Study design; Follow-up[a] | Study population | Categories of nutritional status | Assessment type | Main neurological (N) and neuropsychiatric (NP) symptoms evaluated |
|---|---|---|---|---|---|---|---|
| Farhanah, N. et al., 2022 [45] | Indonesia | To determine persistent symptoms and evaluate quality of life in COVID-19 patients 3 months after discharge | Cohort; 12 weeks after hospital discharge | Hospitalized n = 104 (male = 55) Age: 48.9 (range: 18–65 years) | Excess weight (n = 41) Normal weight (n = 63) | Electronic medical records Telephone interviews Reported symptoms | N: headache, smell disorder, sweating, taste disorder NP: sleep disturbance |
| Fernández-de-Las-Peñas, C. et al, 2021 [54] | Spain | To investigate the association of obesity with long-term post-COVID symptoms in hospitalized COVID-19 survivors | Case-control; 28.8 weeks after hospital discharge | Hospitalized n = 264 (male = 159) Age: 52.0 ± 14.5 years | Obesity (n = 88) Non-obesity (n = 176) | Electronic medical records Telephone interviews List of symptoms Hospital Anxiety and Depression Scale, Pittsburgh Sleep Quality Index (PSQI) | N: Concentration issues, cognitive deficit, headache, memory issues, numbness, sleep disturbance, smell and taste disorders NP: anxiety, depression |
| Gaur, R. et al., 2022 [44] | India | To assess the extent of disability following COVID-19 infection | Cross-sectional; 15.5 weeks after COVID-19 infection | Mixed population[b] n = 97 (male = 61) Age: 48.7 ± 15.6 years | Obesity (n = 32) Overweight (n = 27) Normal weight (n = 38) | In-person interview Measured BMI Questionnaire | N: Headache, vertigo NP: Sleep disturbance |
| Li, Z. et al., 2023[d] [35] | China | To investigate the mental health status of COVID-19 survivors 1 year after discharge and reveal related risk factors | Cross-sectional; 52 weeks (12 months) after hospital discharge | Hospitalized n = 535 (male = 216) Age: 50.8 ± 14.4 years | BMI ≥ 23 (n = 221) BMI 18.5–22.9 (n = 295) | Self-reported survey Measured BMI Questionnaire, General Anxiety Disorder-7 scale, Patient Health Questionnaire (PHQ)-9, PSQI | N: Anxiety, depression NP: Sleep disturbance |
| Miyazato, Y. et al., 2022 [46] | Japan | To explore the factors involved in PCC development in a cohort of patients recovering from COVID-19 at a hospital in Japan | Cross-sectional; 35.5 weeks after symptom onset or COVID-19 diagnosis | Mixed population[b] n = 457 (male = 226) Age: 47 (IQR: 39–55 years) | BMI as continuous variable | Online survey Self-reported BMI Questionnaire with list of symptoms | N: Smell and taste disorders |
| Moy, F.M. et al., 2022 [43] | Malaysia | To investigate mental health status in the form of depression among COVID-19 survivors and its associated factors | Cross-sectional; 27.3 weeks after COVID-19 infection | Mixed population[b] Sample of interest/ study sample n = 567/732 (male = 302) Age: 40.2 ± 10.9 years | Overweight/ obese (n = 315) Normal weight/ underweight (n = 246) | Online survey Reported BMI PHQ-9 | NP: Depression |
| Shang, L. et al., 2021[d] [36] | China | To determine whether obesity has a long-term impact on COVID-19 recovery | Cohort; 46.1 weeks after hospital discharge | Hospitalized n = 118 (male = 48) Age: 53.0 (IQR: 44–61 years) | Obesity (n = 53) Non-obesity (n = 65) | In-person interview Measured BMI Questionnaire | N: Smell disorder NP: sleep disturbance |

*(Continued)*

**Table 1.** (Continued)

| Author, Year | Country | Study aim | Study design; Follow-up[a] | Study population | Categories of nutritional status | Assessment type | Main neurological (N) and neuropsychiatric (NP) symptoms evaluated |
|---|---|---|---|---|---|---|---|
| Sørensen, A.I.V, et al., 2022 [48] | Denmark | To estimate the risk difference between COVID-19 positive and negative individuals; to evaluate the duration of symptoms; to explore the influence of risk factors on persistent symptoms | Cross-sectional; 38 weeks after COVID-19 test. | Mixed populaiton[b] (96% of outpatient) Sample of interest/ study sample $n = 61,002/152,880$ (male = 25,172) Age: 49.0 (IQR: 39–60 years) | Obese ($n = 9,950$) Overweight ($n = 19,264$) Normal weight ($n = 25,285$) | Online survey and electronic records Self-reported BMI Web-based questionnaire | N: Headache, smell disorder, taste disorder, vertigo |
| Van-Wijhe, M. et al., 2022 [49] | Denmark | To investigate the occurrence and risk factors for long-COVID symptoms and health-related quality of life | Cross-sectional; 12 weeks after positive PCR test | Outpatient Sample of interest/ study sample $n = 742/7420$ (male = 245) Age: 48.2 ± 15.0 | Obesity ($n = 148$) Overweight ($n = 226$) Normal weight ($n = 368$) | Electronic patient records and online survey Questionnaire | N: Concentration issues, headache, memory issues, smell disorder, taste disorder, vertigo |
| Vassalini, P. et al., 2021 [52] | Italy | To assess the prevalence of depressive symptoms and related risk factors at 3 months after hospitalization for COVID-19 infection | Cohort 12 weeks after hospital discharge | Hospitalized $n = 115$ (male = 62) Age: 47 (IQR: 48–66 years) | Obesity ($n = 5$) Non-obesity ($n = 110$) | Electronic medical records and telephone interviews PHQ-9 | NP: Depression |
| Whitaker, M. et al, 2022 [50] | England | To estimate symptom prevalence; to investigate co-occurrence of symptoms and assess risk factors for persistence of symptoms | Cross-sectional; 12 weeks after symptom onset | Mixed population [b] Sample of interest/ study sample $n = 78,566/606,434$ (male = 37,600) Age: 80% of sample between 25 and 65 years old | Obesity ($n = 18,892$) Overweight ($n = 27,986$) Normal weight ($n = 30,639$) | Survey (online/ telephone) Self-reported BMI Questionnaire and self-reported symptoms | N: Headache, memory issues, numbness, smell disorder, taste disorder, vertigo NP: sleep disturbance |

BMI: body mass index; PCR: polymerase chain reaction; IQR: interquartile range.

[a]Mean or median of follow-up time, in weeks;

[b]Hospitalized and outpatient;

[c]Argentina, Brazil, Chile, Dominican Republic, Ecuador, Mexico, Panama, Paraguay, and Peru;

[d]Different cut-off values were used to define excess weight and obesity in Asian populations.

for headache ($I^2 = 40.36\%$), taste disorder ($I^2 = 53.46\%$), and vertigo ($I^2 = 59.53\%$). Other analysis showed no significant heterogeneity. Subgroup analyses could not be performed due to an insufficient number of studies evaluating the same outcome by nutritional status.

### Risk-of-bias assessment

We assessed the risk of bias for both evaluated effect measures (frequency and OR). Results for studies reporting the frequency of PCC symptoms are presented in Fig 6; while findings for studies reporting OR are displayed in Supporting Information (S3 Fig). Among the 18 studies evaluated, seven were judged to have a high risk of bias due to self-reported

**Table 2. Frequency of neuropsychiatric symptoms related to post-COVID condition and the risk of their development in included studies.**

| Author, year | Frequency (%) of PCC Symptoms by nutritional status | | | Significantly higher frequency of symptoms in the exposure groups? | Effect of nutritional status on the risk/ or protection from developing a PCC symptom<?Note To TS: Single?> | Exposure increased the risk of symptoms? |
|---|---|---|---|---|---|---|
| | Excess weight *versus* Normal weight | | | (Yes/ No) | aOR (95%CI) | (Yes/ No) |
| | | Exposure groups | Control group | | | |
| Desgranges, F. *et al*., 2022 | Smell and taste disorders | NA | NA | NA | 1.01 (0.61–1.67); *p*=0.96 | No |
| | Headache | | | | 0.79 (0.48–1.75); *p*=0.79 | No |
| | Loss of balance | | | | 1.46 (0.35–6.09); *p*=0.61 | No |
| | Memory issues | | | | 1.79 (0.93–3.44); *p*=0.08 | No |
| | Numbness | | | | 2.74 (0.79–9.50); *p*=0.11 | No |
| | Sleep disturbance | | | | 2.08 (1.03–4.21); *p*=0.04* | Yes |
| Epsi, N.J. *et al*., 2024 | Smell and taste disorder | 0.01 | 0.02 | No | 0.89 (0.37–2.12) | No |
| Li, Z. *et al*., 2023[a] | Anxiety | 16.3 | 13.8; *p*=0.000* | Yes | NA | NA |
| | Depression | 21.7 | 19.0; *p*=0.052* | No | | |
| | Sleep disturbance | 51.1 | 43.3; *p*=0.018* | Yes | | |
| Miyazato, Y. *et al*., 2022 | Smell disorder | NA | NA | NA | 0.94 (0.89–0.99); *p*=0.014* | Yes |
| | Taste disorder | | | | 0.93 (0.88–0.98); *p*=0.012* | Yes |
| Moy, F.M. *et al*., 2022 | Depression | 51.7 | 42.3; p=0.026* | Yes | 1.83 (1.18-2.82) | Yes |
| Van-Wijhe, M. *et al*., 2022 | Concentration issues | NA | NA | NA | 1.01 (0.70-1.47) | No |
| | Headache | | | | 1.09 (0.75-1.58) | No |
| | Memory impairment | | | | 0.94 (0.62-1.42) | No |
| | Taste disorder | | | | 0.85 (0.61-1.32) | No |
| | Smell disorder | | | | 0.72 (0.45-1.05) | No |
| | Vertigo | | | | 0.95 (0.64-1.42) | No |
| | Obesity *versus* Non-obesity | | | (Yes/ No) | OR (95%CI) | (Yes/ No) |
| | | Exposure groups | Control group | | | |
| Chudzik, M.*et al*., 2022 | Smell and taste disorder | 3.1 | 5.1 | No | 0.65 (0.35–1.22); *p*=0.185 | No |
| Epsi, N.J. *et al*., 2024 | Smell and taste disorder | 0.01 | 0.02 | No | 0.76 (.29–1.99) | No |

*(Continued)*

**Table 2.** (Continued)

| Author, year | Frequency (%) of PCC Symptoms by nutritional status | | | Significantly higher frequency of symptoms in the exposure groups? | Effect of nutritional status on the risk/ or protection from developing a PCC symptom<?Note To TS: Single?> | Exposure increased the risk of symptoms? |
|---|---|---|---|---|---|---|
| | Excess weight *versus* Normal weight | | | (Yes/ No) | aOR (95%CI) | (Yes/ No) |
| | | Exposure groups | Control group | | | |
| Fernández-de-las-Peñas, C. *et al, 2021* | Ageusia | 8.0 | 6.8 | Yes | NA | NA |
| | Anosmia | 2.3 | 6.8 | No | NA | NA |
| | Anxiety | 15.9 | 9.7 | Yes | 1.75 (0.82–3.72); *p*=0.146 | No |
| | Cognitive deficit | 9.1 | 6.8 | Yes | NA | NA |
| | Concentration issues | 13.6 | 9.1 | Yes | NA | NA |
| | Depression | 13.6 | 15.9 | No | 0.83 (0.40–1.73); *p*=0.628 | No |
| | Headache | 8.0 | 5.7 | Yes | NA | NA |
| | Memory impairment | 18.2 | 14.8 | Yes | NA | NA |
| | Sleep disturbance | 45.5 | 25.6 | Yes | 2.10 (1.13–3.83); *p*=0.020* | Yes |
| Shang, L. *et al.,* 2021[a] | Sleep disturbance | 39.6 | 44.6; *p*=0.590 | No | NA | NA |
| | Smell disorder | 7.5 | 7.7; *p*=0.970 | No | | |
| Van-Wijhe, M. *et al.,* 2022 | Concentration issues | NA | NA | NA | 0.96 (0.62-1.49) | No |
| | Headache | | | | 1.06 (0.69-1.65) | No |
| | Memory impairment | | | | 0.92 (0.56-1.50) | No |
| | Taste disorder | | | | 1.00 (0.65-1.55) | No |
| | Smell disorder | | | | 0.83 (0.53-1.55) | No |
| | Vertigo | | | | 0.90 (0.56-1.45) | No |

* Statistically significant; NA: non-available information; excess-weight group (BMI ≥ 25 kg/m$^2$); normal-weight group (BMI < 25 kg/m$^2$); obesity group (BMI ≥ 30 kg/m$^2$); non-obesity group (BMI < 30 kg/m$^2$). [a] Different cut-off values were used to define excess weight and obesity in Asian populations (excess weight ≥ 23 kg/m$^2$; obesity ≥25 kg/m$^2$).

exposure data [43,46–48,50,55,58]. The ROBINS-E tool indicated that no further evaluation of a study was required for studies with inappropriate measurement of exposure or outcome; therefore, this tool was applied only to the remaining 11 studies [35,36,44,45,49,51–54,56,57].

The domain most affecting the assessment was control for confounders (Domain 1 [D1]). Few studies employed appropriate designs (e.g., randomization or matching) [49,54,57] or analytical methods (e.g., stratification) [35,51–53]; only one study had an overall low risk of bias [53]. Concerns were also noted regarding exposure measurement (Domain 2 [D2]), particularly when data were obtained from electronic medical records without reporting time-related information [54].

Findings from the GRADE assessment are presented in Table 3 and supporting information (S7 Table). The quality of evidence was downgraded from high to very low for headache, taste disorder, and vertigo (with excess weight as the exposure group) and for anxiety, taste disorder and sleep disturbance (with obesity as exposure group) due to high risk of

bias, imprecision of results, and publication bias. For the other outcomes, the quality of evidence was rated as low, having been downgraded from high due to risk of bias and publication bias. Depression and cognitive issues among individuals with obesity and the control group were rated as moderate certainty of evidence, downgraded from high due to publication bias.

## Discussion

Although it has been suggested that individuals with increased BMI are at higher risk of PCC, the associations between EW and specific PCC related neurological and neuropsychiatric symptoms remained unclear. Our review reveals that EW is significantly associated with a range of persistent and PCC symptoms, including headache, vertigo, smell and taste disorder, sleep disturbance and depression. These findings suggest that EW might contribute to the development of these symptoms that persist for more than 12 weeks after COVID-19 onset. Moreover, this study is opportune as we transition into the post-pandemic period facing the challenges of managing the co-occurrence of pandemics, including overweight/obesity, mental health issues, and the burden of PCC.

Increased BMI has been identified as a determinant of adverse outcomes during both the acute and chronic phases of COVID-19 [18,27,29]. This contrasts with the role of sex in the disease course since the pattern of acute phase symptoms by sex is distinct from that of PCC, which tends to affect more females [20,59]. Additionally, the chronic phase of COVID-19's has been variously defined, with some studies indicating symptoms lasting for at least 4 weeks [12] and others extending to more than 12 weeks [5,6]. Differentiating these time frames is crucial to distinguish between acute illness and potential sequelae of irreversible tissue damage, which may present with varying degrees of impairment [60]. In our review, we used the cut-off criterion to ensure that we focused on long-term or chronic outcomes of COVID-19. The mean follow-up time from illness onset in the included studies was 25.8 weeks (ranging from 12 to 52 weeks). A large systematic review [2] evaluated the prevalence of persistent symptoms among COVID-19 survivors at different follow-up periods. It identified sleep disorder and concentration difficulties as the most common symptoms at 3–6 months (24%, 95% CI: 8%–44%; 22%, 95% CI: 15%–31%, respectively), with sleep disorder being most prevalent at over 12 months. However, unlike our results, this review had a high degree of between-study heterogeneity and did not include BMI in subgroups analysis.

Long-term neurological symptoms significantly decrease the quality of life for individuals, a situation that might be exacerbated by a pre-exiting comorbidities [61]. Moreover, symptoms related to comorbidities often overlap with those of PCC, leading individuals with obesity to attribute their symptoms to their metabolic disease rather than to persistent manifestations of COVID-19 [62,63]. Additionally, overweight individuals also report health complaints, including headache and sleeplessness, associated to their subclinical disease state also known as suboptimal health status, which can be exacerbated by PCC symptoms [22]. The co-occurrence of these complex conditions can have deleterious effects, impairing people's daily functioning and increasing the demand on healthcare systems. Our results highlight a significantly positive association for persistent headache, vertigo, numbness and taste disorder in individuals with EW or obesity. These results are consistent with recent study demonstrating that obesity increased the risk for memory disorders and neurological cluster of symptoms (headaches, expectoration, myalgias, fatigue, and taste and smell disorders) among COVID-19 survivors [64]. This underscores the role of excessive body weight (fat deposits) in PCC. It is worth noting that patients with PCC have reported that head pain worsens with physical exercise while a decrease in physical activity coupled with increase in sedentary behaviour are generally known to contribute to weight gain [65,66]. Moreover, taste disorders may influence unhealthy weight-related behaviours, such as consuming more palatable, ultra-processed foods high in salt, sugar, and additives. These foods can further impair gustatory function and exacerbate unhealthy symptoms [67,68]. Vertigo which impairs daily functioning by increasing the risk of falls may also be associated with depression [69,70]. To date, the diagnosis of PCC relies on clinical judgement as no defined biomarkers currently exist to confirm the condition [8]. Furthermore, a widely agreed-upon treatment for PCC has yet to emerge. This highlights gaps in scientific knowledge

a) Anxiety

| Study | Excess weight | | Normal weight | | RR with 95% CI | Weight (%) |
|---|---|---|---|---|---|---|
| | Symptom | No symptom | Symptom | No symptom | | |
| Blumel, 2022 | 4 | 161 | 1 | 138 | 3.37 [ 0.38, 29.80] | 3.45 |
| Li, Z. et al., 2023[a] | 36 | 185 | 41 | 254 | 1.17 [ 0.78, 1.77] | 96.55 |
| Overall | | | | | 1.22 [ 0.81, 1.82] | |

Heterogeneity: $\tau^2 = 0.00$, $I^2 = 0.00\%$, $H^2 = 1.00$

Test of $\theta_i = \theta_j$: Q(1) = 0.87, p = 0.35

Test of $\theta = 0$: z = 0.94, p = 0.34

Random-effects DerSimonian–Laird model

b) Depression

| Study | Excess weight | | Control | | RR with 95% CI | Weight (%) |
|---|---|---|---|---|---|---|
| | Symptom | No symptom | Symptom | No symptom | | |
| Li, Z. et al., 2023[a] | 48 | 173 | 56 | 239 | 1.14 [ 0.81, 1.61] | 21.66 |
| Moy, 2021 | 163 | 152 | 104 | 142 | 1.22 [ 1.02, 1.47] | 78.34 |
| Overall | | | | | 1.21 [ 1.03, 1.42] | |

Heterogeneity: $\tau^2 = 0.00$, $I^2 = 0.00\%$, $H^2 = 1.00$

Test of $\theta_i = \theta_j$: Q(1) = 0.12, p = 0.73

Test of $\theta = 0$: z = 2.30, p = 0.02

Random-effects DerSimonian–Laird model

c) Sleep disturbance

| Study | Excess weight | | Normal weight | | RR with 95% CI | Weight (%) |
|---|---|---|---|---|---|---|
| | Symptom | No symptom | Symptom | No symptom | | |
| Alkwai, H.M. et al., 2022 | 3 | 29 | 14 | 167 | 1.21 [ 0.37, 3.98] | 1.04 |
| Gaur, R. et al., 2022 | 9 | 50 | 2 | 36 | 2.90 [ 0.66, 12.69] | 0.68 |
| Blümel, J.E. et al., 2022 | 4 | 161 | 3 | 136 | 1.12 [ 0.26, 4.93] | 0.68 |
| Farhanah, N. et al., 2022 | 0 | 41 | 2 | 61 | 0.30 [ 0.02, 6.19] | 0.16 |
| Li, Z. et al., 2023[a] | 113 | 108 | 128 | 167 | 1.18 [ 0.98, 1.42] | 28.58 |
| Whitaker, M. et al., 2022 | 3,711 | 43,167 | 1,780 | 28,859 | 1.36 [ 1.29, 1.44] | 68.85 |
| Overall | | | | | 1.31 [ 1.16, 1.48] | |

Heterogeneity: $\tau^2 = 0.00$, $I^2 = 17.83\%$, $H^2 = 1.22$

Test of $\theta_i = \theta_j$: Q(5) = 4.27, p = 0.51

Test of $\theta = 0$: z = 4.29, p = 0.00

Random-effects REML model

**Fig 3. Forest plots of excess weight and risk ratio (RR) for neuropsychiatric symptoms.** [a] Applied a different BMI cut-off (Excess weight > 23kg/m2)..

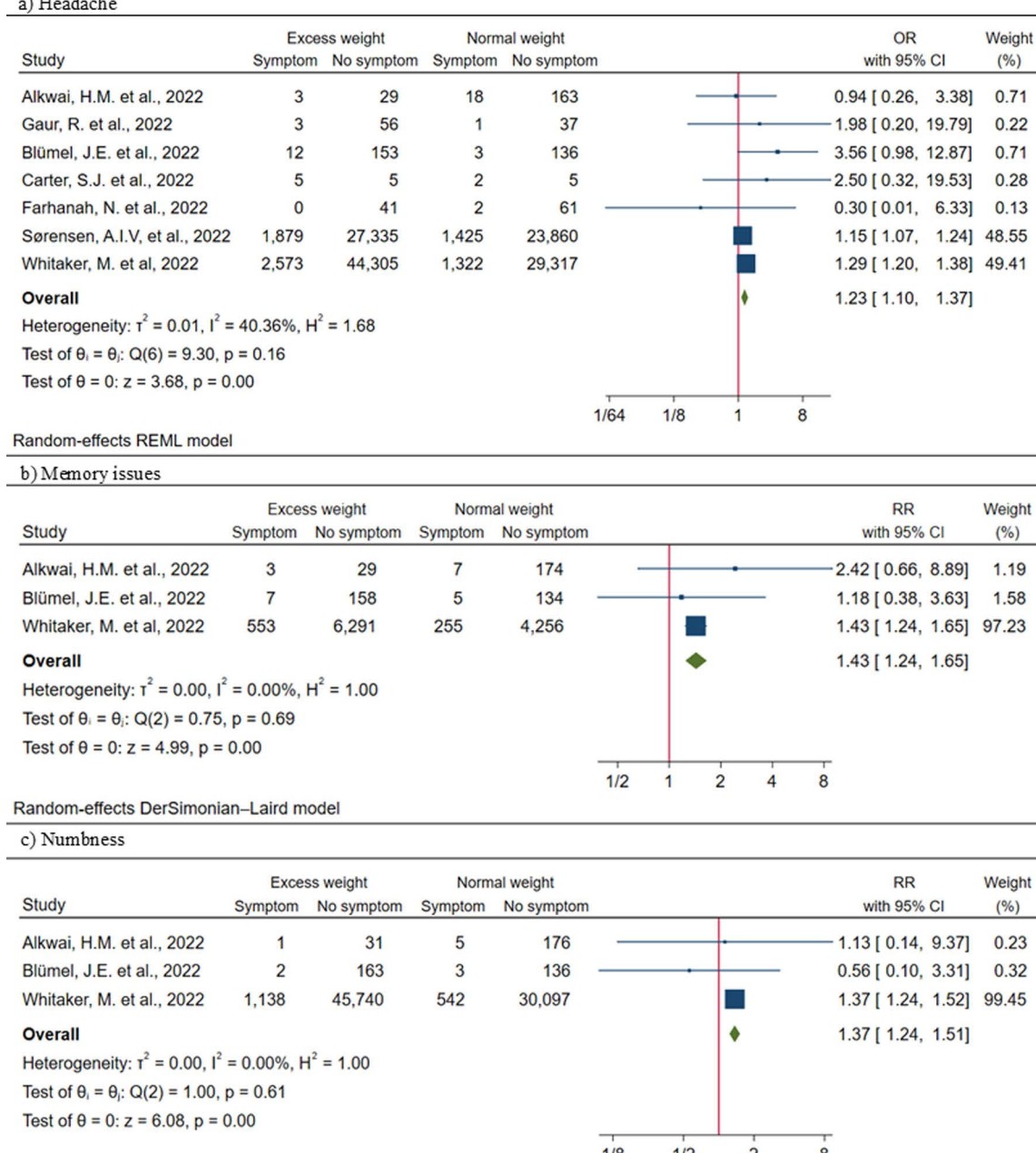

**Fig 4. A.** Forest plots of excess weight and pooled results (RR and OR) for neurological symptoms. **B.** Forest plots of excess weight and pooled results (RR and OR) for neurological symptoms. **C.** Forest plots of excess weight and pooled results (RR and OR) for neurological symptoms.

and the urgent need for government agencies, especially of low and middle-income countries, to develop evidence-based clinical practice guidelines and training programs for health-care workers, especially in primary care services to improve diagnosis and enable more comprehensive patients care [71]. Therefore, healthcare systems and public policies should

d) Smell disorder

| Study | Excess weight | | Normal weight | | | RR with 95% CI | Weight (%) |
|---|---|---|---|---|---|---|---|
| | Symptom | No symptom | Symptom | No symptom | | | |
| Alkwai, H.M. et al., 2022 | 6 | 26 | 21 | 160 | | 1.62 [ 0.71, 3.69] | 0.20 |
| Blümel, J.E. et al., 2022 | 13 | 152 | 16 | 123 | | 0.68 [ 0.34, 1.37] | 0.29 |
| Farhanah, N. et al., 2022 | 0 | 40 | 1 | 63 | | 0.53 [ 0.02, 12.67] | 0.01 |
| Sørensen, A.I.V, et al., 2022 | 3,195 | 26,019 | 2,781 | 22,504 | | 0.99 [ 0.95, 1.04] | 60.35 |
| Whitaker, M et al., 2022 | 2,640 | 44,238 | 1,671 | 28,968 | | 1.03 [ 0.97, 1.10] | 39.14 |
| **Overall** | | | | | | 1.01 [ 0.97, 1.05] | |

Heterogeneity: $\tau^2 = 0.00$, $I^2 = 0.00\%$, $H^2 = 1.00$
Test of $\theta_i = \theta_j$: Q(4) = 3.54, p = 0.47
Test of $\theta = 0$: z = 0.47, p = 0.64

1/32  1/8  1/2  2  8

Random-effects DerSimonian–Laird model

e) Smell and taste disroder

| Study | Excess weight | | Normal weight | | | RR with 95% CI | Weight (%) |
|---|---|---|---|---|---|---|---|
| | Symptom | No symptom | Symptom | No symptom | | | |
| Alkwai, H.M. et al., 2022 | 6 | 26 | 21 | 160 | | 1.62 [ 0.71, 3.69] | 0.20 |
| Blümel, J.E. et al., 2022 | 13 | 152 | 16 | 123 | | 0.68 [ 0.34, 1.37] | 0.29 |
| Farhanah, N. et al., 2022 | 0 | 40 | 1 | 63 | | 0.53 [ 0.02, 12.67] | 0.01 |
| Sørensen, A.I.V, et al., 2022 | 3,195 | 26,019 | 2,781 | 22,504 | | 0.99 [ 0.95, 1.04] | 60.35 |
| Whitaker, M et al., 2022 | 2,640 | 44,238 | 1,671 | 28,968 | | 1.03 [ 0.97, 1.10] | 39.14 |
| **Overall** | | | | | | 1.01 [ 0.97, 1.05] | |

Heterogeneity: $\tau^2 = 0.00$, $I^2 = 0.00\%$, $H^2 = 1.00$
Test of $\theta_i = \theta_j$: Q(4) = 3.54, p = 0.47
Test of $\theta = 0$: z = 0.47, p = 0.64

1/32  1/8  1/2  2  8

Random-effects DerSimonian–Laird model

**Fig 4.** Continued.

focus on multidisciplinary rehabilitation services to address the long-term impacts of COVID-19 on survivors, specifically through concomitant personalized management of weight, neurological and neuropsychiatric issues. Cognitive behaviour therapy (CBT) and programs that combine both physical and mental rehabilitation may improve cognitive function and patient recovery and are among as the most effective interventions recently recommended for PCC treatment [72]. Social and occupational support should also be addressed to contribute to compassionate and effective care of patients [71].

The high prevalence of persistent complex concurrent symptoms may create a vicious circle between physical and neuropsychiatric symptoms, increasing the risk for depression [73–78]. A significant positive association between obesity and depressed mood has been observed across multiple COVID-19 periods, similar to findings in other coronavirus outbreaks [11,16,66]. During the COVID-19 pandemic, a systematic review of longitudinal studies identified obesity/overweight as a risk factor for depression (pooled RR = 1.2; 95% CI: 1.11–1.31). However, no significant association was found when obesity/overweight was self-reported (pooled RR = 1.03; 95% CI: 0.99–1.26) [79]. In this study, we observed a significantly positive association for persistent depression with EW. This findings aligns with Aminian *et al.* [80] who reported that the need for diagnostic test to assess neuropsychiatric problems (a *proxy* for symptoms) was significantly higher in individuals with increased BMI compared to those with normal weight. Although our pooled RR for EW and depression included data from two studies conducted with Asian populations [35,43] featuring different BMI assessments (measured and self-reported) and BMI cut-offs (excess weight defined as BMI ≥ 23 kg/m² and obesity as BMI ≥ 25 kg/m²), both studies used a validated depression scale (PHQ-9), and the results were not heterogeneous.

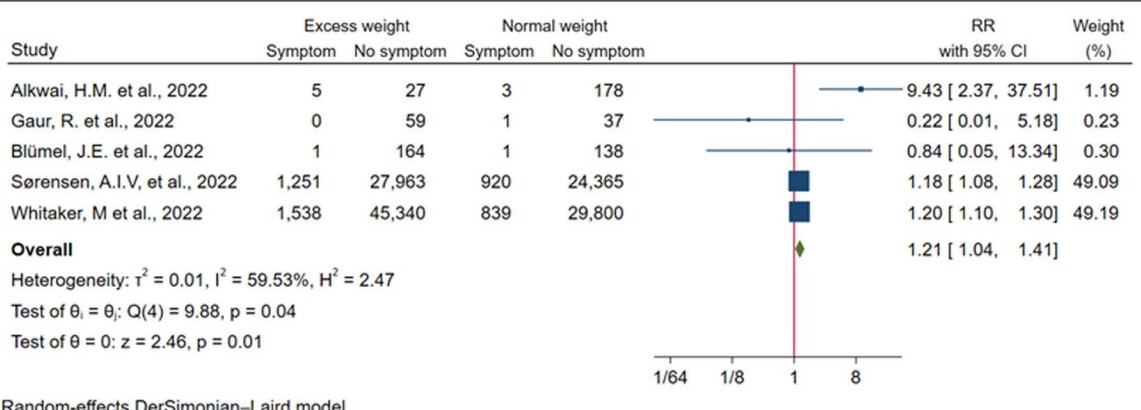

f) Taste disorder

| Study | Excess weight Symptom | Excess weight No symptom | Control Symptom | Control No symptom | RR with 95% CI | Weight (%) |
|---|---|---|---|---|---|---|
| Alkwai, H.M. et al., 2022 | 1 | 31 | 14 | 167 | 0.40 [ 0.06, 2.97] | 0.26 |
| Blümel, J.E. et al., 2022 | 3 | 162 | 2 | 137 | 1.26 [ 0.21, 7.46] | 0.33 |
| Farhanah, N. et al., 2022 | 1 | 40 | 2 | 61 | 0.77 [ 0.07, 8.20] | 0.18 |
| Sørensen, A.I.V, et al., 2022 | 2,616 | 26,598 | 2,160 | 23,125 | 1.05 [ 0.99, 1.11] | 51.45 |
| Whitaker, M et al., 2022 | 2,293 | 44,585 | 1,267 | 29,372 | 1.18 [ 1.11, 1.26] | 47.78 |
| **Overall** | | | | | 1.11 [ 1.00, 1.23] | |

Heterogeneity: $\tau^2 = 0.00$, $I^2 = 53.46\%$, $H^2 = 2.15$

Test of $\theta_i = \theta_j$: Q(4) = 8.60, p = 0.07

Test of $\theta = 0$: z = 1.98, p = 0.05

Random-effects DerSimonian–Laird model

g) Vertigo

| Study | Excess weight Symptom | Excess weight No symptom | Normal weight Symptom | Normal weight No symptom | RR with 95% CI | Weight (%) |
|---|---|---|---|---|---|---|
| Alkwai, H.M. et al., 2022 | 5 | 27 | 3 | 178 | 9.43 [ 2.37, 37.51] | 1.19 |
| Gaur, R. et al., 2022 | 0 | 59 | 1 | 37 | 0.22 [ 0.01, 5.18] | 0.23 |
| Blümel, J.E. et al., 2022 | 1 | 164 | 1 | 138 | 0.84 [ 0.05, 13.34] | 0.30 |
| Sørensen, A.I.V, et al., 2022 | 1,251 | 27,963 | 920 | 24,365 | 1.18 [ 1.08, 1.28] | 49.09 |
| Whitaker, M et al., 2022 | 1,538 | 45,340 | 839 | 29,800 | 1.20 [ 1.10, 1.30] | 49.19 |
| **Overall** | | | | | 1.21 [ 1.04, 1.41] | |

Heterogeneity: $\tau^2 = 0.01$, $I^2 = 59.53\%$, $H^2 = 2.47$

Test of $\theta_i = \theta_j$: Q(4) = 9.88, p = 0.04

Test of $\theta = 0$: z = 2.46, p = 0.01

Random-effects DerSimonian–Laird model

**Fig 4.** Continued.

It remains uncertain whether neurological and neuropsychiatric manifestations related to PCC are directly attributable to the virus itself or if they develop indirectly (such as, through an immune response or medical therapy). These symptoms may also involve both the central and peripheral nervous systems [16]. Additionally, the role of EW in the progression of COVID-19 is not fully understood, though it may be linked to an exaggerated inflammatory response or pre-existing genetic factors that these conditions share [81–83]. The adipose tissue plays a role in SARS-CoV-2 entry and deposition, and it may serve as a reservoir for virus spread. The nature of adipocytes and elevated fatty acid levels may enhance virus replication and contribute to delayed viral clearance, which is associated with persistent symptoms [84]. EW is also a recognized risk factor for various chronic conditions and the number of pre-existing comorbidities has been associated to the development of PCC symptoms up to two years after COVID-19 onset, in both hospitalized and non-hospitalized patients [85]. Our results underscore the importance of identifying at-risk individuals and highlight the need for timely personalized interventions for COVID-19 survivors. These findings could inform the development of predictive and preventive management plans for future waves of SARS-CoV-2 or other epidemics.

Our quality assessment results indicate that the included studies suffer from methodological issues, particularly in measuring exposure. Studies that classified individuals' nutritional status based on self-reported data were found to have a high risk of bias, which contributed to downgrade in the overall quality of evidence for the evaluated symptoms. Self-reported data are subject to recall bias, social desirability bias, and variations in individual interpretation of symptoms,

a) Cognitive issues

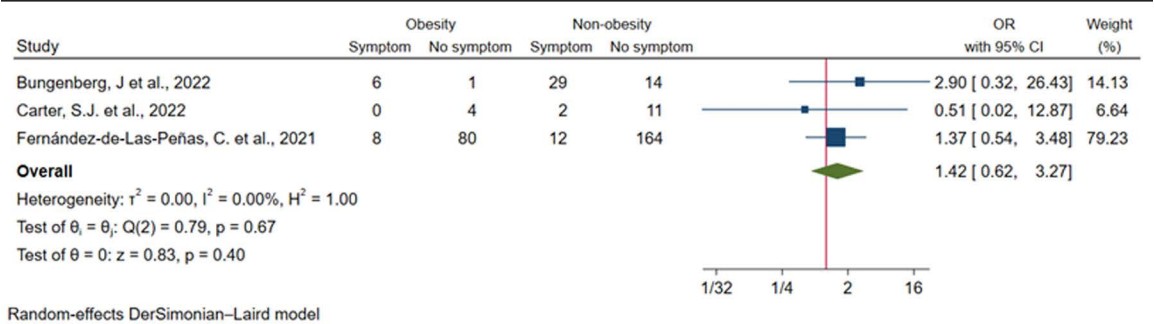

b) Headache

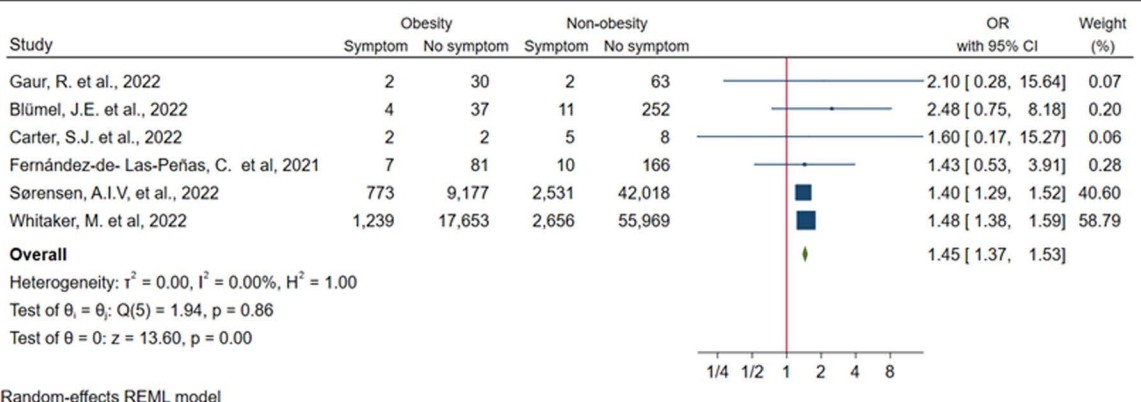

c) Memory issues

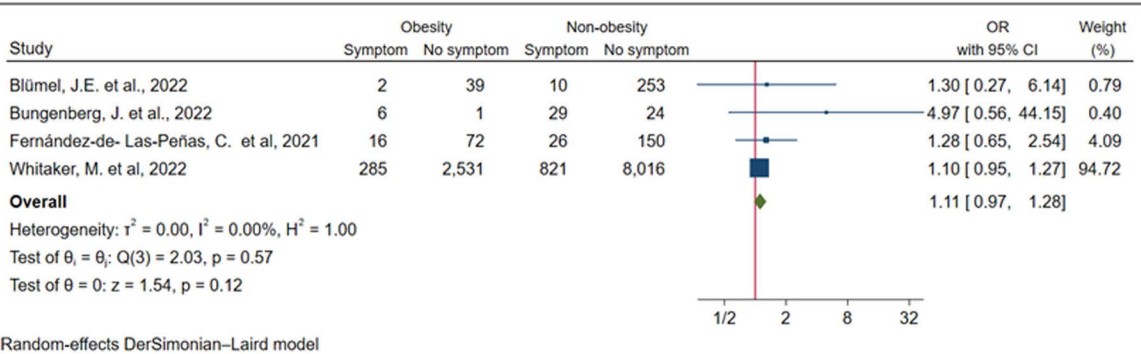

**Fig 5. A. Forest plot of obesity and pooled results (RR and OR) for neurological symptoms. B. Forest plot of obesity and pooled results (RR and OR) for neurological symptoms.** [a] Applied a different BMI cut-off (obesity >25 kg/m²). **C. Forest plot of obesity and pooled results (RR and OR) for neurological symptoms.**

which may affect the accuracy of the findings. Moreover, during the height of the pandemic, online health surveys were often used when other research methods were infeasible [86]. However, this design might not adequately capture exposure and could therefore influence the precision of outcomes. Future research should aim to complement self-reported data with objective clinical assessments to enhance the validity of the results and address these potential biases. Issues

d) Numbness

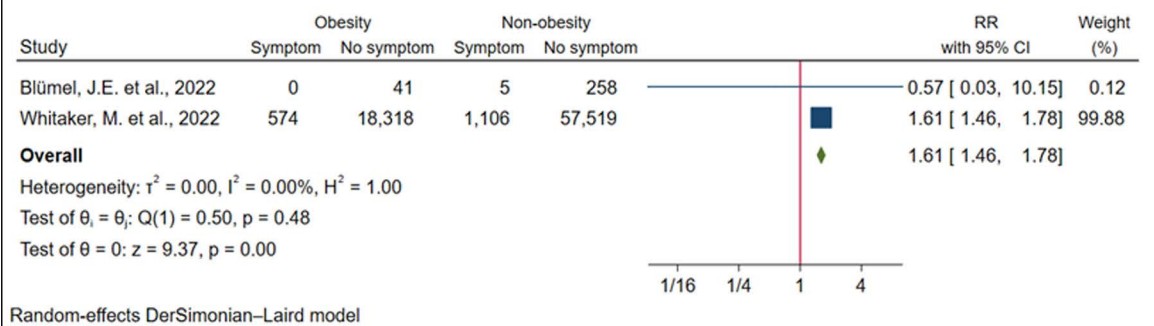

e) Smell disorder

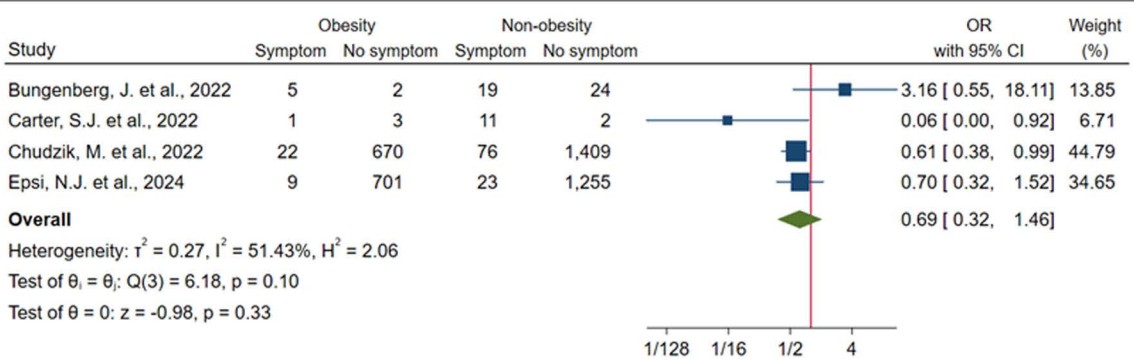

f) Smell and taste disroder

Fig 5. Continued.

could also arise in data from medical records as time-related information was not reported. While a few included studies adopted recommended strategies to reduce bias in survey design, which enhanced the rigor of their findings, controlling confounding variables remained a significant concern. In particular, inadequate control of factors such as the severity of COVID-19 acute-phase, sex, previous existing comorbidities and duration of symptoms was frequently observed. Moreover, significant issues also arise from studies relying on self-reported symptoms without using structured questionnaire or validated scales. Although screening tools have been proposed to identify individuals with PCC they often do not consider neuropsychiatric symptoms [87] or key cognitive symptoms [88] that significantly affect individuals' quality of

g) Taste disorder

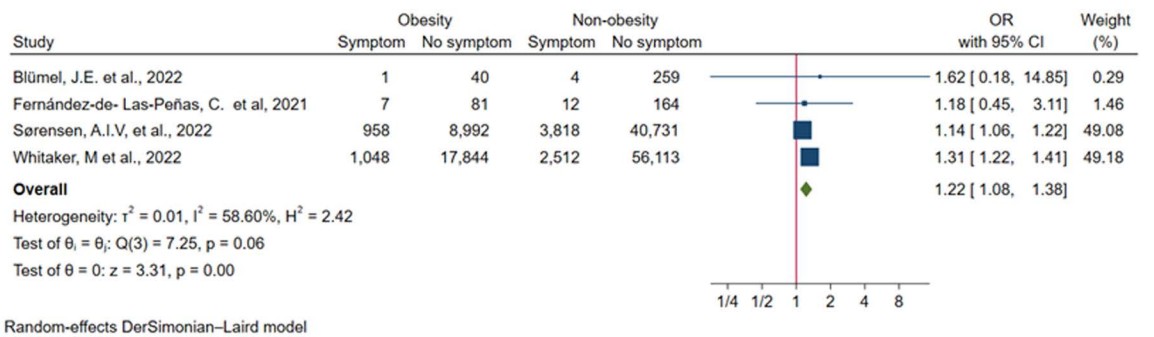

h) Vertigo

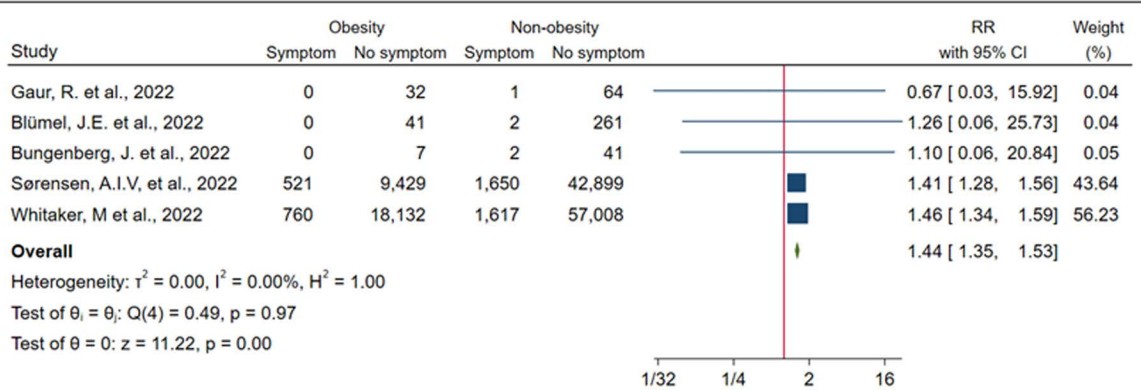

**Fig 5.** Continued.

life. The increased heterogeneity observed among studies evaluating anxiety, headache, taste disorder and vertigo has led to imprecision in the results. Additionally, the limited number of available studies to be included in the meta-analysis made it impossible to assess publication bias, further contributing to the downgrading of the overall quality of evidence for these outcomes from high to very low. The inability to assess publication bias also reduced the certainty of evidence of all evaluated outcomes and highlight the significant knowledge gap regarding risk factors associated with PCC. Very low and low overall certainty of evidence suggests that the estimated effects may differ substantially from the true effects and highlights the need for caution when drawing conclusions and applying the results to clinical practice. Therefore, future research is urgently needed to investigate the combined effects of these chronic conditions.

It is important to acknowledge certain limitations when interpreting the findings of our research. Most of the included studies were cross-sectional which limits the ability to determine cause and effect relationships. The inconsistency in naming PCC symptoms among studies resulted in a relatively low number of studies reporting the same symptom which also precluded subgroup comparisons between hospitalized and outpatient populations. Moreover, the varying symptoms' terminologies applied and the use of clusters of symptoms analysis among eligible studies might have led to underestimation of the results. We included studies that used non-validated questionnaires to assess symptoms, as it was necessary given that PCC is an emerging condition and validated screening tools have yet to be developed. Although the cut-off for study inclusion regarding time to follow-up (i.e., > 12 weeks) allowed us to track long-term outcomes from COVID-19 and address a gap in the literature, the follow-up period ranged from 12 to 52 weeks post-infection, the varying risk of PCC symptoms throughout different follow-up periods was not investigated in subgroup analysis due to limited number of studies. The

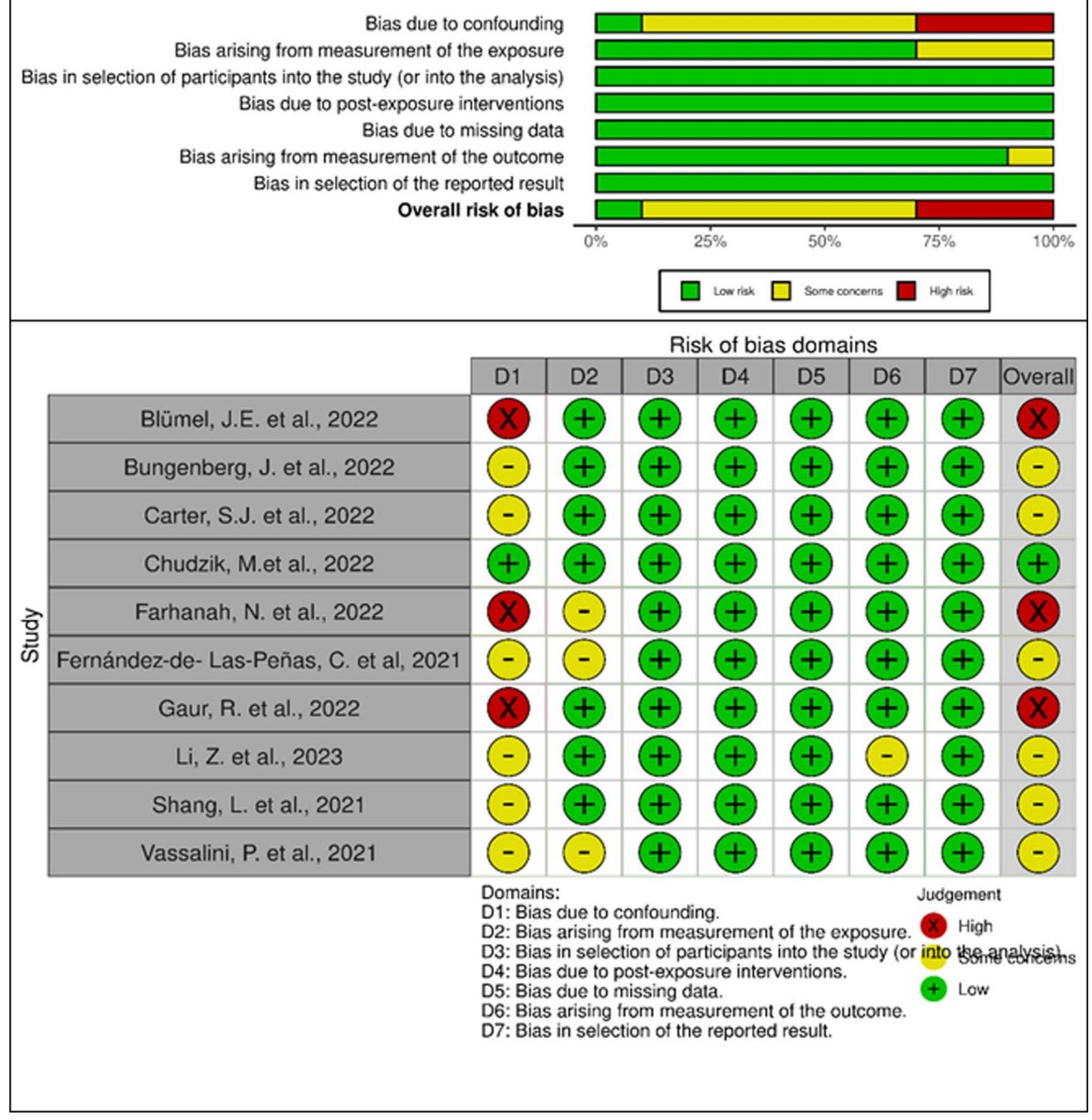

**Fig 6. Summary plot and traffic light plot illustrating the risk of bias in the included studies.**

available data did not allow for reliable subgroup analyses based on demographic variables (e.g., age, sex, follow-up period) or COVID-19 severity as well. Additionally, the inclusion of both obesity and non-obesity as comparisons groups led to the interchange of overweight individuals between exposure and control groups. Nevertheless, defining EW and obesity as exposure groups was crucial given the syndemic context. This approach enabled the evaluation of the risk of developing persistent symptoms among different levels of excessive fat and highlighted that both suboptimal health status (overweight) and illness (obesity) predisposed individuals to PCC-related neuro-symptoms. However, the inherent differences among these groups should be considered in health, disease prevention and diagnoses. Furthermore, the issues related to body fat assessment should be noted as BMI remains the most widely used screening tool for measuring adiposity. Although it

**Table 3. Overall certainty of the evidence assessment for the association of neurological and neuropsychiatric symptoms of Post-Covid-19 Condition and excess weight.**

| Certainty assessment | | | | | | | № of patients | | Effect | | Certainty |
|---|---|---|---|---|---|---|---|---|---|---|---|
| № of studies | Study design | Risk of bias | Incon-sistency | Indirect-ness | Impre-cision | Other considerations | Excess weight | normal weight | Relative (95% CI) | Absolute (95% CI) | |
| **Depression (follow-up: range 27 weeks to 52 weeks; assessed with: Scales)** | | | | | | | | | | | |
| 2 | non-randomised studies | seri-ous[a] | not serious | not serious[a] | not serious | publication bias strongly suspected[b] | 211/536 (39.4%) | 160/541 (29.6%) | RR 1.21 (1.03 to 1.42) | 62 more per 1000 (from 9 more to 124 more) | ◼◻◻◻ Low[a,b] |
| **Sleep disturbance (follow-up: range 12 weeks to 52 weeks; assessed with: Questionnaire and scales)** | | | | | | | | | | | |
| 6 | non-randomised studies | seri-ous[a] | not serious | not serious | not serious | publication bias strongly suspected[b] | 3861/47449 (8.1%) | 1958/31420 (6.2%) | RR 1.31 (1.16 to 1.48) | 19 more per 1000 (from 10 more to 30 more) | ◼◻◻◻ Low[a,b] |
| **Headache (follow-up: range 12 weeks to 387 weeks; assessed with: Questionnaire and self-reported)** | | | | | | | | | | | |
| 7 | non-randomised studies | seri-ous[a] | not serious | not serious | seri-ous[c] | publication bias strongly suspected[b] | 7248 cases 132.751 controls 4475/76399 exposed 2773/56352 unexposed | | RR 1.15 (1.07 to 1.22) | 0 fewer per 1000 (from 0 fewer to 0 fewer) | ◼◻◻◻ Very low[a,b,c] |
| **Memory issues (follow-up: range 12 weeks to 32 weeks; assessed with: Questionnaire)** | | | | | | | | | | | |
| 3 | non-randomised studies | seri-ous[a] | not serious | not serious | not serious | publication bias strongly suspected[b] | 563/7041 (8.0%) | 267/4831 (5.5%) | RR 1.43 (1.24 to 1.65) | 24 more per 1000 (from 13 more to 36 more) | ◼◻◻◻ Low[a,b] |
| **Numbness (follow-up: range 12 weeks to 32 weeks; assessed with: Questionnaire)** | | | | | | | | | | | |
| 3 | non-randomised studies | seri-ous[a] | not serious | not serious | not serious | publication bias strongly suspected[b] | 1141/47075 (2.4%) | 550/30959 (1.8%) | RR 1.37 (1.24 to 1.51) | 7 more per 1000 (from 4 more to 9 more) | ◼◻◻◻ Low[a,b] |
| **Taste disorder (follow-up: range 12 weeks to 38 weeks; assessed with: Questionnaire and self-reported)** | | | | | | | | | | | |
| 5 | non-randomised studies | seri-ous[a] | not serious | not serious | seri-ous[c] | publication bias strongly suspected[b] | 4914/76330 (6.4%) | 3445/56307 (6.1%) | RR 1.11 (1.00 to 1.23) | 7 more per 1000 (from 0 fewer to 14 more) | ◼◻◻◻ Very low[a,b,c] |
| **Vertigo (follow-up: range 12 weeks to 38 weeks; assessed with: Questionnaire and self-reported)** | | | | | | | | | | | |
| 5 | non-randomised studies | seri-ous[a] | not serious | not serious | seri-ous[c] | publication bias strongly suspected[b] | 2795/76348 (3.7%) | 1764/56282 (3.1%) | RR 1.21 (1.04 to 1.41) | 7 more per 1000 (from 1 more to 13 more) | ◼◻◻◻ Very low[a,b,c] |

CI: confidence interval; OR: odds ratio; RR: risk ratio;

a. The risk of bias of included studies is related to the measurement of exposure (self-reported) or the control of confounders, particularly regarding the severity of COVID-19 acute phase;

b. There are less than 10 studies included in the metanalyses therefore we were unable to evaluate publication bias;

c. Metanalyses showed moderate heterogeneity among included studies.

is a simple and non-invasive measure often related to the gold standard fat assessment BMI does not accounts for age, nor does it for ethnic backgrounds and the types or distribution of adipose tissue [89]. In addition, high visceral fat exhibits more angiotensin-converting enzyme-2 (ACE2) levels than subcutaneous adipose tissue, thereby it is more susceptible to SARS-CoV-2 entry and replication, resulting in higher viral load [84,90]. Thus, future research should focus on longitudinal evaluation of COVID-19 survivors incorporating objective assessment of symptoms and adiposity.

Despite these limitations, our review has several scientific and clinical strengths. Notably, the number of investigated outcomes allowed for a pioneering evaluation of the association of EW with both the physical and psychological long-term manifestations of COVID-19. Additionally, the comprehensive definition of PCC that we applied during our article search (to account for varied terminology), along with the inclusion of each neurological and neuropsychiatric symptom specified in our search strategy, ensured a thorough screening process. We also underscored the importance of addressing EW as a global health concern, given its interplay with neurological and neuropsychiatric manifestations, and infectious diseases such as COVID-19. Moreover, this systematic review contributes to the growing body of literature on post-COVID-19 condition by detailing the range of neurological and neuropsychiatric symptoms experienced by adults with both post-COVID-19 condition and excess weight. By revealing these association, our study underscores the significant physical and mental health burden faced by COVID-19 survivors with excess weight and obesity, offering critical insight to guide rehabilitation efforts. Therefore, the management of EW should be considered in the treatment of neurological and neuro-psychiatric symptoms due to the significant impact that these combined conditions have on individuals' health.

## Conclusions

Our systematic review and meta-analysis demonstrate that EW is significantly associated with post-COVID-19 neurological and neuropsychiatric symptoms, including headache, memory issues, numbness, smell and taste disorders, vertigo, depression, and sleep disturbance. These findings underscore for the importance of developing multidisciplinary rehabilitative strategies tailored to individual needs to improve care management and support the overall health of affected individuals. Our results provide evidence-based guidance for healthcare professionals and policymakers in managing PCC and contribute to ongoing global efforts to understand its underlying mechanisms, epidemiology, and identification.

## Supporting information

**S1 Table. Database search strategy.**
(DOC)

**S2 Table. Description of identified records and reasons for exclusion.**
(XLSX)

**S3 Table. Data extracted from included studies (n = 18)**[a,b] . [a] All data was extracted by DBR and LOM; [b] Due to the number of columns the table was split in two to enable the presentation of all data extracted; PCR: polymerase chain reaction; NA: Data non-available.
(DOCX)

**S4 Table. Data extracted from included studies for meta-analysis.** PTotal: Total population; POB: Population with obesity; PNOB: Non-obesity population; POW: Population with overweight; PEW: population with excess weight; PEUT: Eutrophic population; NSymptomOB: *n* of individuals with obesity that reported the symptom; NSymptomNOB: *n* of non-obesity individuals that reported the symptom; NSymptomOW: *n* of individuals with overweight that reported the symptom; NSymptomEUT: *n* of eutrophic individuals that reported the symptom; NSymptomEW: *n* of individuals with excess weight that reported the symptom; NOSymptomOB: *n* of individuals with obesity that did not report the symptom; NOSymptomNOB: *n* of non-obesity individuals that did not report the symptom; NOSymptomOW: *n* of individuals with

overweight that did not report the symptom; NOSymptomEUT: *n* of eutrophic individuals that did not report the symptom; NOSymptomEW: *n* of individuals with excess weight that did not report the symptom.
(DOCX)

**S5 Table. Author contact details and requested data**[a]. [a]**All corresponding authors of included studies were contacted by e-mail by Debora Barbosa Ronca (DBR –** deboraronca@gmail.com**); BMI: body mass index; PCR: polymerase chain reaction.**
(DOCX)

**S6 Table. Frequency of neurological and neuropsychiatric symptoms of Post-COVID-19 Condition from included studies that did not test for statistical differences according to nutritional status.** Excess weight group (BMI ≥ 25 kg/m$^2$); NW: Normal Weight group (BMI < 25 kg/m$^2$); Obesity group (BMI ≥ 30 kg/m$^2$); NOb: Non-obesity group (BMI < 30 kg/m$^2$).
(DOCX)

**S7 Table. Assessment of the overall certainty of the evidence for the association of neurological and neuropsychiatric symptoms of PCC between exposure and control groups.** A) Excess weight versus normal weight; B) Obesity versus non-obesity group. CI: confidence interval; OR: odds ratio; RR: risk ratio; a. The risk of bias of included studies is related to the measurement of exposure (self-reported) or the control of confounders, particularly regarding the severity of COVID-19 acute phase; b. There are less than 10 studies included in the metanalyses therefore we were unable to evaluate publication bias;.c. Metanalysis showed substantial heterogeneity among included studies; d. Metanalysis showed moderate heterogeneity among included studies.
(DOCX)

**S1 Fig. Forest-plots of the association of excess weight and the risk of neurological and neuropsychiatric symptoms.**
(DOCX)

**S2 Fig. Forest-plots of obesity and odds ratio (OR) for neuropsychiatric symptoms.**
(TIF)

**S3 Fig. Traffic light plots of risk of bias of included studies that reported the risk of developing neuro-symptoms, assessed using the Robbins-e tool.**
(DOCX)

## Acknowledgments

We would like to thank the researchers who shared information and enabled more precise results in this study. We would also like to acknowledge students and collaborators of the PENSA research group at the University of Brasília and the Centre for Precision Health at Edith Cowan University for their support.

## Author contributions

**Conceptualization:** Débora Barbosa Ronca, Larissa Otaviano Mesquita, Kênia Mara Baiocchi de Carvalho.

**Formal analysis:** Débora Barbosa Ronca, Larissa Otaviano Mesquita, Ana Cláudia Morais Godoy Figueiredo.

**Funding acquisition:** Débora Barbosa Ronca.

**Investigation:** Larissa Otaviano Mesquita, Dryelle Oliveira.

**Project administration:** Kênia Mara Baiocchi de Carvalho.

**Supervision:** Ana Cláudia Morais Godoy Figueiredo, Kênia Mara Baiocchi de Carvalho.

**Writing – original draft:** Débora Barbosa Ronca, Kênia Mara Baiocchi de Carvalho.

**Writing – review & editing:** Larissa Otaviano Mesquita, Ana Cláudia Morais Godoy Figueiredo, Jun Wen, Manshu Song, Kênia Mara Baiocchi de Carvalho.

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
