## [Decision Letter · Decision Letter 0]

9 Dec 2024

PONE-D-24-52453Excess weight increases the risk for neurological and neuropsychiatric symptoms in post-COVID-19 condition: A systematic review and meta-analysisPLOS ONE

Dear Dr. Ronca,

Thank you for submitting your manuscript to PLOS ONE. After careful consideration, we feel that it has merit but does not fully meet PLOS ONE’s publication criteria as it currently stands. Therefore, we invite you to submit a revised version of the manuscript that addresses the points raised during the review process.

We look forward to receiving your revised manuscript.

Kind regards,

Dong Keon Yon, MD, FACAAI, FAAAAI

Academic Editor

PLOS ONE

2. As required by our policy on Data Availability, please ensure your manuscript or supplementary information includes the following:

3. Please include a caption for figure 3.

Additional Editor Comments:

Please address the excellent comments from the reviewers.

Reviewers' comments:

Reviewer's Responses to Questions

**Comments to the Author**

1. Is the manuscript technically sound, and do the data support the conclusions?

Reviewer #1: Yes

Reviewer #2: Yes

2. Has the statistical analysis been performed appropriately and rigorously? 

Reviewer #1: No

Reviewer #2: Yes

3. Have the authors made all data underlying the findings in their manuscript fully available?

Reviewer #1: Yes

Reviewer #2: No

4. Is the manuscript presented in an intelligible fashion and written in standard English?

Reviewer #1: Yes

Reviewer #2: No

5. Review Comments to the Author

Reviewer #1: This systematic review and meta-analysis studied the association between excess weight and post-acute neuropsychiatric sequelae of COVID-19. Despite conducting a rigorous research study, drawing the cause-and-effect relationship between excess weight and post-acute neuropsychiatric sequelae of COVID-19 is statistically inappropriate due to the inclusion of cross-sectional studies. I also suggest using odds ratios instead of relative risks. Other comments are as follows:

- Page 3 line 61: "Although PCC has no uniform definition in the literature". PCC has a universal definition according to the National Academies of Sciences, Engineering, and Medicine (N Engl J Med. 2024 Nov 7;391(18):1746-1753.)

- Page 9 line 214: "Statistical heterogeneity was assessed using the I2 statistic". How was substantial heterogeneity defined? This will affect your decision on downgrading the certainty of evidence due to the inconsistency of the results.

Reviewer #2: Major Comments

1. Provide a detailed discussion on the clinical applicability of the findings to post-COVID-19 care for patients with excess weight.

2. Justify the exclusion of studies focusing on comorbidities and their potential influence on the outcomes.

3. Elaborate on the impact of high risk of bias and heterogeneity, as highlighted in the ROBINS-E and GRADE assessments.

4. Perform subgroup analyses for demographic variables (e.g., age, sex) and COVID-19 severity to refine interpretations.

5. Address methodological limitations of included studies, particularly regarding exposure measurement and confounder control.

6. Clarify the choice of random-effects models and provide details on the handling of missing data in the meta-analyses.

7. Integrate findings with the broader literature on post-COVID-19 conditions and obesity.

Minor Comments

1. Ensure figure captions are precise and consistent with the content.

2. Clearly define terms such as "suboptimal health condition" with appropriate citations.

3. Streamline data presentation in tables and shift less critical details to supplementary materials.

4. Expand on ethical considerations, particularly for studies relying on self-reported data.

5. Update references to include recent, high-impact studies relevant to the research question.

6. Ensure the funding statement explicitly addresses the minimization of funder influence.

Figures and Data

1. Standardize formatting across all forest plots to enhance clarity and interpretability.

2. Include additional plots (e.g., funnel plots, sensitivity analyses) to assess and mitigate publication bias.

6. PLOS authors have the option to publish the peer review history of their article (what does this mean? ). If published, this will include your full peer review and any attached files.

**Do you want your identity to be public for this peer review?** For information about this choice, including consent withdrawal, please see our Privacy Policy .

Reviewer #1: No

Reviewer #2: No

---

## [Author Response · Author response to Decision Letter 0]

17 Feb 2025

On January 28th 2025

1. Please ensure that you refer to Figure 3 in your text as, if accepted, production will need this reference to link the reader to the figure

An appropriate caption has been included for Fig 3 (Lines 308-309). The figure 3 is also referred in the text (line 300).

2. Please provide a numbered table of all those 10125 studies identified in the literature search, including those that were excluded from the analyses

A new file encompassing all studies identified in the literature search, including those excluded from the analyses, has been added to the manuscript as supplementary material (S2 Table). This is also referenced in the main text (see Line 154). We also included this information in our updated cover letter, to highlight our response to the editor’s request after the revised submission. It is worth mentioning that the included file (.xlxs format) has two sheets, named “Databases” and “Citation searching”. The “Databases” sheet encompasses 10,122 records identified from databases and the “Citation searching” sheet encompasses 30 studies identified from citation searching. For all the excluded studies, the reasons for exclusion were described. Moreover, an updated PRISMA flowchart was uploaded as figure 1 due to a minor typing error identified (instead of 10,125 records screened from databases, 10,122 records were screened – as indicated in the abstract (line 38) and results sessions (line 234).

On December 09th 2024:

Response: The financial disclosure has been included in the updated cover letter as follows: “This research received a grant from the Research Support Program at the School of Health Sciences, Brasília, Brazil, funded by the Health Sciences Teaching and Research Foundation (Grant and Acceptance Term No. 5/2020 – FEPECS/DE) (DBR). This study was also partially financed by the Coordenação de Aperfeiçoamento de Pessoal de Nível Superior, Brasil – Finance Code 001 (DBR). MS is supported by the Western Australian Future Health Research and Innovation Fund (Grant ID WANMA/Ideas2023-24/10). KMBC is supported by National Council for Scientific and Technological Development CNPq (Grant n. 302740/2022-8). The funders had no role in the study design, data collection and analysis, decision to publish, or preparation of the manuscript.”

Response: The manuscript has been revised to meet PLOS ONE's style requirements, including file naming conventions. We hope this amended version meets the journal's expectations.

2. As required by our policy on Data Availability, please ensure your manuscript or supplementary information includes the following:

Response: A new file encompassing all studies identified in the literature search, including those excluded from the analyses, has been added to the manuscript as supplementary material (S2 Table). This is also referenced in the main text (see Line 154). We also included this information in our updated cover letter, to highlight our response to the editor’s request after the revised submission. It is worth mentioning that the included file (.xlxs format) has two sheets, named “Databases” and “Citation searching”. The “Databases” sheet encompasses 10,122 records identified from databases and the “Citation searching” sheet encompasses 30 studies identified from citation searching. For all the excluded studies, the reasons for exclusion were described. Moreover, an updated PRISMA flowchart was uploaded as figure 1 due to a minor typing error identified (instead of 10,125 records screened from databases, 10,122 records were screened – as indicated in the abstract (line 38) and results sessions (line 234).

Response: A file listing all studies identified in the literature search, along with the reasons for exclusion, has been added to the manuscript (S2 Table and Line 154).

Response: All included studies have been previously published or are available as preprints.

- Name of data extractors and date of data extraction;

- Confirmation that the study was eligible to be included in the review;

Response: Supplementary files (S3 Table and S4 Table) containing the data extracted from the included studies, along with the names of data extractors, extraction dates, eligibility confirmation, and all relevant data, have been included with the manuscript (S3 Table, S4 Table, and Line 188).

Response: A supplementary file (S5 Table) with details on the type and dates of data obtained from another source has been included in the manuscript (S5 Table and Line 189).

Response: The reviewer is right about the importance of presenting the risk of bias and GRADE for each study, including all items of the instruments and for each outcome. We applied the ROBBINS-E tool to assess the risk of bias for the included studies, taking into account both frequency and odds ratio measures of effect. The completed risk of bias assessment is shown in Fig. 6 and S3 Fig. Additionally, we used the GRADE tool to assess the quality/certainty of evidence for all investigated outcomes (symptoms), as presented in Table 3 and S7 Table. It is worth mentioning that Table 3 presents the certainty of evidence for the main outcomes evaluated by GRADE, according to Cochrane recommendations, which emphasize patient-important outcomes (Cochrane Handbook 2012, Section 5.4a).

Response: This systematic review included studies that reported the frequency (n, %) of persistent neurological and neuropsychiatric symptoms associated with post-Covid-19 condition (PCC), categorized by nutritional status among COVID-19 survivors. Studies were eligible if they reported data on one or more symptoms of the evaluated outcomes. The meta-analyses were conducted using all available data for each symptom, with separate analyses for each outcome. Corresponding authors were contacted to share data on symptom frequencies by nutritional status (S6 Table). For studies that did not report data for a specific symptom, that study was excluded from the corresponding meta-analysis, ensuring that missing data was not an issue. As such, no imputation of missing data was necessary. Additionally, due to the limited number of studies reporting the same outcome (e.g., anxiety, headache) across exposure/control groups (obesity vs. non-obesity, excess weight vs. normal weight), we were unable to perform the Egger’s test to evaluate publication bias. (Lines 229-230). Finally, considering the differences in symptoms terminology among eligible studies and issues regarding how missing data were handled we included as limitation of our study that this inconsistency in naming PCC symptoms might have led to underestimation of results due to limited data of persistent symptoms according to nutritional status (Lines 513-515).

Response: We ensure that we have provided all the underlying data required for publication in this journal.

3. Please include a caption for figure 3.

Answer: An appropriate caption has been included for Fig 3 (Lines 308-309). The figure 3 is also referred in the text (line 300).

Additional Editor Comments:

Please address the excellent comments from the reviewers.

Response: Indeed, the reviewers' comments were excellent, which contributed to the improvement of the manuscript. We are grateful for that.

Reviewers' comments:

Reviewer's Responses to Questions

Comments to the Author

5. Review Comments to the Author

Reviewer #1: This systematic review and meta-analysis studied the association between excess weight and post-acute neuropsychiatric sequelae of COVID-19. Despite conducting a rigorous research study, drawing the cause-and-effect relationship between excess weight and post-acute neuropsychiatric sequelae of COVID-19 is statistically inappropriate due to the inclusion of cross-sectional studies. I also suggest using odds ratios instead of relative risks. Other comments are as follows:

Answer: We appreciate the reviewer’s comments. As suggested, odds ratios (ORs) are commonly reported in cross-sectional studies when logistic regression is used to assess the association between risk factors and outcome while controlling for confounders. When the outcome is rare (<10%), the OR and risk ratio (RR) are approximately equal, and the OR can be used to approximate the RR (allowing the OR to serve as an approximation of the RR) (Ref.: Wilber ST & Fu R., 2010). We acknowledge as a limitation of our study that the inclusion of cross-sectional studies limits the ability to establish causal relationships (Lines 509-510). However, our data reveals a significant association between excess weight and persistent neurological and neuropsychiatric symptoms in post-COVID-19 Condition. In response to the reviewer’s suggestion, we have revised the manuscript to replace terms related to “risk” with “is associated with/association” for greater clarity. The title of the manuscript was also updated to avoid misunderstandings. Additionally, to address the concern about statistical appropriateness, we reviewed all pooled analysis involving case-control included studies (Carter SJ, et al., 2022 and Fernandes-De-Las-Penas C, et al. 2021) and recalculated the results as Odds Ratios (ORs) instead of Risk Ratios (RRs). Specifically, pooled ORs were calculated for the following comparisons: obesity versus non-obesity groups for cognitive impairment, headache, memory issues, smell disorder, taste disorder, smell and taste disorder, anxiety, depression and sleep disturbance; and excess weight versus normal weight for headache, smell and taste disorder. We have also updated the Methods section to clarify our approach: “We performed meta-analytic calculations using STATA software (SE/17). Pooled risk ratios (RRs) with their 95% confidence intervals (CIs) were computed from the raw data of included cohort and cross-sectional studies, while pooled odds ratios (OR) were computed for reported symptoms identified in case-control studies” (Lines 217-220).

References:

1. Wilber ST, Fu R. Risk ratios and odds ratios for common events in cross-sectional and cohort studies. Acad Emerg Med. 2010 Jun;17(6):649-51. doi: 10.1111/j.1553-2712.2010.00773.x. PMID: 20624147.

2. Fernández‐de‐las‐Peñas C, Torres‐Macho J, Elvira‐Martínez CM, Molina‐Trigueros LJ, Sebastián‐Viana T, Hernández‐Barrera V. Obesity is associated with a greater number of long‐term post‐COVID symptoms and poor sleep quality: A multicentre case‐control study. Int J Clin Pract 2021; 75: e14917.

3. Carter SJ, Baranauskas MN, Raglin JS, Pescosolido BA, Perry BL. Functional Status, Mood State, and Physical Activity Among Women With Post-Acute COVID-19 Syndrome. Int J Public Health 2022; 67: 1604589.

- Page 3 line 61: "Although PCC has no uniform definition in the literature". PCC has a universal definition according to the National Academies of Sciences, Engineering, and Medicine (N Engl J Med. 2024 Nov 7;391(18):1746-1753.)

Response: We appreciate the reviewer’s comment. While a universal definition of PCC was proposed by the National Academies of Sciences, Engineering, and Medicine in November 2024, various broad and non-specific definitions were used in the literature to capture the diverse manifestations of PCC. For instance, the Centers for Disease Control and Prevention (CDC) defined as any signs, symptoms, or conditions persisting for at least four weeks after infection, whereas the World Health Organization (WHO) defined it as the continuation or development of new, otherwise unexplained symptoms three months after the initial SARS-CoV-2 infection. All included studies followed different criteria and definitions of PCC, as the universal definition by the National Academies of Sciences, Engineering, and Medicine was published only in November 2024. Thus, we updated our manuscript and included this important reference in our introduction sections. Moreover, considering the available references during the time of our study and the proposed universal definition, we changed the phrase “Although PCC has no uniform definition in the literature” to “Although PCC has varying definition in the literature” (line 61) and also included information regarding varying terminologies in lines 103-107.

References:

WHO. A clinical case definition of post-COVID-19 condition by a Delphi consensus - The Lancet Infectious Diseases, https://www.thelancet.com/journals/laninf/article/PIIS1473-3099(21)00703-9/fulltext (accessed 15 October 2023).

- Page 9 line 214: "Statistical heterogeneity was assessed using the I2 statistic". How was substantial heterogeneity defined? This will affect your decision on downgrading the certainty of evidence due to the inconsistency of the results.

Response: We defined substantial heterogeneity based on the thresholds recommended by Cochrane: 0%-40% (no important heterogeneity), 30%-60% (moderate heterogeneity), 50%-90% (substantial heterogeneity), and 75-100% (considerable heterogeneity). This information was included in the Methods section of the revised manuscript (Line 225-228). To determine the degree of heterogeneity, we considered the I2 value, visual inspection of forest plot, the confidence interval, the direction of the effects, and the clinical and methodological differences among the included studies.

References:

Higgins J, Thomas J, Chandler J, Cumpston M, Li T, Page M, et al. Cochrane Handbook for Systematic Reviews of Interventions, https://training.cochrane.org/handbook/current (accessed 12 June 2024).

Reviewer #2: Major Comments

1. Provide a detailed discussion on the clinical applicability of the findings to post-COVID-19 care for patients with excess weight.

Response: We have expanded the Discussion section of the manuscript to include a detailed analysis of the clinical applicability

---

## [Decision Letter · Decision Letter 1]

12 Mar 2025

PONE-D-24-52453R1Excess weight is associated with neurological and neuropsychiatric symptoms in post-COVID-19 condition: A systematic review and meta-analysisPLOS ONE

Dear Dr. Ronca,

Thank you for submitting your manuscript to PLOS ONE. After careful consideration, we feel that it has merit but does not fully meet PLOS ONE’s publication criteria as it currently stands. Therefore, we invite you to submit a revised version of the manuscript that addresses the points raised during the review process.

We look forward to receiving your revised manuscript.

Kind regards,

Dong Keon Yon, MD, FACAAI, FAAAAI

Academic Editor

PLOS ONE

Journal Requirements:

Additional Editor Comments:

Please see my minor comments

#1. Studies published up to July 2023 were searched independently across eight electronic databases (PubMed ..... Please describe it) to evaluate the risk of developing...

#2. post-COVID infection -> post-COVID-19 condition or SARS-CoV-2 infection

#3. I am simply curious—what is the reason for the omission of this particular paper?

https://pubmed.ncbi.nlm.nih.gov/38918517/

#4. Please describe GRADE system in method section in more dtail.

Reviewers' comments:

Reviewer's Responses to Questions

**Comments to the Author**

1. If the authors have adequately addressed your comments raised in a previous round of review and you feel that this manuscript is now acceptable for publication, you may indicate that here to bypass the “Comments to the Author” section, enter your conflict of interest statement in the “Confidential to Editor” section, and submit your "Accept" recommendation.

Reviewer #1: All comments have been addressed

2. Is the manuscript technically sound, and do the data support the conclusions?

Reviewer #1: Yes

3. Has the statistical analysis been performed appropriately and rigorously? 

Reviewer #1: Yes

4. Have the authors made all data underlying the findings in their manuscript fully available?

Reviewer #1: Yes

5. Is the manuscript presented in an intelligible fashion and written in standard English?

Reviewer #1: Yes

6. Review Comments to the Author

Reviewer #1: (No Response)

7. PLOS authors have the option to publish the peer review history of their article (what does this mean? ). If published, this will include your full peer review and any attached files.

**Do you want your identity to be public for this peer review?** For information about this choice, including consent withdrawal, please see our Privacy Policy .

Reviewer #1: No

---

## [Author Response · Author response to Decision Letter 1]

21 Mar 2025

Response: We carefully reviewed our reference list and made the necessary corrections. We identified five studies with published errata and/or corrections. In all cases, the authors confirmed that these were minor errors that did not affect the studies’ findings or conclusions. These studies were published in prestigious scientific journals (The New England Journal of Medicine, Nature Reviews Microbiology, The Lancet and PLOS ONE, General Hospital Psychiatry). Accordingly, we updated our reference list to reflect this information, as detailed below. Both the original and updated reference lists have been included in the cover letter, with the corrected references highlighted. We confirm that we did not cite any retracted papers.

References updated:

1) Blumenthal D et al., 2020.

Original: Blumenthal D, Fowler EJ, Abrams M, Collins SR. Covid-19 — Implications for the Health Care System. N Engl J Med 2020; 383: 1483–1488.

Updated: Blumenthal D, Fowler EJ, Abrams M, Collins SR. Covid-19 — Implications for the Health Care System. [published correction appears in N Engl J Med. 2020 Oct 22;383(17):1698. N Engl J Med 2020; 383: 1483–1488.

2) Davis HE et al., 2023

Original: Davis HE, McCorkell L, Vogel JM, Topol EJ. Long COVID: major findings, mechanisms and recommendations. Nat Rev Microbiol 2023; 21: 133–146.

Correction reference included: Davis HE, McCorkell L, Vogel JM, Topol EJ. Author Correction: Long COVID: major findings, mechanisms and recommendations. Nat Rev Microbiol 2023; 21: 408–408.

3) Swinburn BA, et al., 2019

Original: Swinburn BA, Kraak VI, Allender S, Atkins VJ, Baker PI, Bogard JR, et al. The Global Syndemic of Obesity, Undernutrition, and Climate Change: The Lancet Commission report. Lancet Lond Engl. 2019 393(10173):791–846. DOI: 10.1016/S0140-6736(18)32822-8.

Errata reference included: Errata in: Department of Error. Lancet Lond Engl. 2019 393(10173):746. DOI: 10.1016/S0140-6736(19)30384-8.

4) Hassan NM, et al., 2023

Original: Hassan NM, Salim HS, Amaran S, Yunus NI, Yusof NA, Daud N, et al. Prevalence of mental health problems among children with long COVID: A systematic review and meta-analysis. PLOS ONE. 2023 18(5):e0282538. DOI: 10.1371/journal.pone.0282538.

Correction reference included: PLOS ONE Staff. Correction: Prevalence of mental health problems among children with long COVID: A systematic review and meta-analysis. PLoS One. 2023 Dec 14;18(12):e0296160. doi: 10.1371/journal.pone.0296160. Erratum for: PLoS One. 2023 May 17;18(5):e0282538. doi: 10.1371/journal.pone.0282538. PMID: 38096220; PMCID: PMC10721052.

5) van der Feltz-Cornelis C, et al., 2024.

Original: van der Feltz-Cornelis C, Turk F, Sweetman J, Khunti K, Gabbay M, Shepherd J, et al. Prevalence of mental health conditions and brain fog in people with long COVID: A systematic review and meta-analysis. Gen Hosp Psychiatry. 2024 88:10–22. DOI: 10.1016/j.genhosppsych.2024.02.009.

Correction reference included: van der Feltz-Cornelis CM, Turk F, Sweetman J, Khunti K, Gabbay M, Shepherd J, et al. Corrigendum to ‘Prevalence of mental health conditions and brain fog in people with long COVID: A systematic review and meta-analysis’ [General Hospital Psychiatry volume 88 (2024)10-22 10.1016/j.genhosppsych.2024.02.009]. Gen Hosp Psychiatry. 2025 92:112. DOI: 10.1016/j.genhosppsych.2024.09.006.

Additional Editor Comments:

Please see my minor comments

#1. Studies published up to July 2023 were searched independently across eight electronic databases (PubMed ..... Please describe it) to evaluate the risk of developing...

Response: This sentence was included in the original version of the manuscript but was updated in the revised version. It was replaced with the following: “We conducted a comprehensive search of eight databases, including PubMed and Embase, for studies published up to July 2023.” To meet the editor’s requirements, we explicitly listed the eight databases in the abstract (lines 32–34): “We conducted a comprehensive search of eight databases (PubMed, Embase, SCOPUS, Web of Science, VHL, Google Scholar, ProQuest, and medRxiv) for studies published up to July 2023.” The discrepancy may have occurred because both the original and revised manuscripts were included in the revised submission, and we did not withdraw the original version. We apologize for any confusion and appreciate the opportunity to clarify. We hope that the updated sentence aligns with the editor’s comments.

#2. post-COVID infection -> post-COVID-19 condition or SARS-CoV-2 infection

Response: We change the sentence accordingly (line 73).

#3. I am simply curious—what is the reason for the omission of this particular paper?

https://pubmed.ncbi.nlm.nih.gov/38918517/

Response: Although this well-designed study aimed to explore the risk of neuropsychiatric complications following a COVID-19 diagnosis using nationally representative data (South Korean and Japanese populations), it did not meet the established eligibility criteria of our systematic review. The study analysed clusters of neuropsychiatric events according to nutritional status (Body Mass Index – BMI) rather than assessing specific symptoms individually, as required by our criteria (Supplementary Information, Tables S7 and S27). For instance, Tables S7 and S27 presented a stratified analysis of the long-term risk of neuropsychiatric events by BMI but did not report the risk for each of the 13 evaluated categories separately. To meet our eligibility criteria, the study would have needed to provide data for each category stratified by BMI, similar to the subgroup analyses in Tables S9 and S10, which assessed long-term hazard ratios (HR) for neuropsychiatric disorders according to COVID-19 severity and vaccination status. Due to variations in the symptom clusters analysed across studies, we excluded studies that examined clusters of symptoms according to nutritional status. Furthermore, the title/abstract of the study “Short- and long-term neuropsychiatric outcomes in long COVID in South Korea and Japan” did not mention “excess weight,” “overweight,” “obesity,” or “body mass index.” These terms represented Exposure in our predefined PECO framework and were included in our search strategy. Their absence from the title/abstract prevented the identification and retrieval of the study through our search. The exclusion criteria of our systematic review are detailed in the Methods section (lines 137–146), and the search strategy is outlined in the Methods section (lines 121–127) and Supporting Information (S1).

#4. Please describe GRADE system in method section in more detail.

Response: We appreciate the editor’s comments and have updated the Methods section of our study to provide a more detailed description of the GRADE system (lines 200-215).

---

## [Editor Report · Decision Letter 2]

24 Mar 2025

Excess weight is associated with neurological and neuropsychiatric symptoms in post-COVID-19 condition: A systematic review and meta-analysis

PONE-D-24-52453R2

Dear Dr. Ronca,

We’re pleased to inform you that your manuscript has been judged scientifically suitable for publication and will be formally accepted for publication once it meets all outstanding technical requirements.

Kind regards,

Dong Keon Yon, MD, FACAAI, FAAAAI

Academic Editor

PLOS ONE

Additional Editor Comments (optional):

This is an excellent paper.
---

## [Editor Report · Acceptance letter]

PONE-D-24-52453R2

PLOS ONE

Dear Dr. Ronca,

I'm pleased to inform you that your manuscript has been deemed suitable for publication in PLOS ONE. Congratulations! Your manuscript is now being handed over to our production team.

Kind regards,

on behalf of

Dr. Dong Keon Yon

Academic Editor

PLOS ONE